# Epiphytic diatom community structure and richness is determined by macroalgal host and location in the South Shetland Islands (Antarctica)

Andrea M. Burfeid-Castellanos[1]*, Rafael P. Martín-Martín[2], Michael Kloster[1], Carlos Angulo-Preckler[3], Conxita Avila[4], Bánk Beszteri[1]

1 Universität Duisburg-Essen, Phycology, Essen, Germany, 2 Laboratory of Botany, Faculty of Pharmacy and Food Science, University of Barcelona (UB), Barcelona, Spain, 3 Norwegian College of Fishery Science, UiT, The Arctic University of Norway, Tromsø, Norway, 4 Faculty of Biology, Department of Evolutionary Biology, Ecology, and Environmental Sciences, and Institute of Biodiversity Research (IrBIO), Universitat de Barcelona, Barcelona, Catalonia

* andrea.burfeid-castellanos@uni-due.de

**Data Availability Statement:** All image files are available from the PANGAEA database (https://doi.pangaea.de/10.1594/PANGAEA.925913). All code

## Abstract

The marine waters around the South Shetland Islands are paramount in the primary production of this Antarctic ecosystem. With the increasing effects of climate change and the annual retreat of the ice shelf, the importance of macroalgae and their diatom epiphytes in primary production also increases. The relationships and interactions between these organisms have scarcely been studied in Antarctica, and even less in the volcanic ecosystem of Deception Island, which can be seen as a natural proxy of climate change in Antarctica because of its vulcanism, and the open marine system of Livingston Island. In this study we investigated the composition of the diatom communities in the context of their macroalgal hosts and different environmental factors. We used a non-acidic method for diatom digestion, followed by slidescanning and diatom identification by manual annotation through a web-browser-based image annotation platform. Epiphytic diatom species richness was higher on Deception Island as a whole, whereas individual macroalgal specimens harboured richer diatom assemblages on Livingston Island. We hypothesize this a possible result of a higher diversity of ecological niches in the unique volcanic environment of Deception Island. Overall, our study revealed higher species richness and diversity than previous studies of macroalgae-inhabiting diatoms in Antarctica, which could however be the result of the different preparation methodologies used in the different studies, rather than an indication of a higher species richness on Deception Island and Livingston Island than other Antarctic localities.

## Introduction

On Antarctic coasts, marine macro- and microalgae in ice and benthos are the main primary producers [1]. When free of shelf-ice, these benthic producers can account for up to 90% of

and datafiles are accessible at DRYAD database (https://doi.org/10.5061/dryad.ngf1vhhsm).

**Funding:** Samples were collected in the frame of the DISTANTCOM (Diversity and Structure of benthic Antarctic communities, CTM2013-42667/ANT) and BLUEBIO (Bioactive marine natural products in our environmentally changing planet, CTM2016-78901) grants funded by the Spanish Government (CA). The Deutscher Akademischer Austauschdienst (DAAD) funded ABC with a short-term grant (57442045, grant number 91673491). We also acknowledge support by the Open Access Publication Fund of the University Duisburg-Essen.

**Competing interests:** The authors have declared that no competing interests exist.

the total net primary production of the ecosystem [1]. The surface of macroalgal hosts also serves as habitat for benthic microalgae (mainly diatoms) in the Antarctic and Subantarctic regions. Although macroalgae cannot be interpreted either as synonymous with or part of plants in the systematic sense, these macroalgae-inhabiting diatom assemblages are usually referred to as "epiphytic" in the literature [2–4]. This association provides a basis for complex ecological interactions between diatoms and macroalgal hosts, which are only partially understood [5]. It is known that epiphytic diatoms can facilitate the adherence of other organisms to any substrate [6]. Epiphytic diatoms and other biofouling organisms can also reduce light intensity available to the host algae [7]. Interactions with surface-inhabiting diatoms can influence the performance of macro- and microalgae species to acclimate or adapt to new ecosystem pressures like climate change [8,9] or invasive species from lower latitudes [10,11].

Previous studies on macroalgae- inhabiting diatoms have focussed on the taxonomic composition and ecology of the epiphytic diatom flora on macroalgae around Antarctica [2,3,5,12–15] or terrestrial habitats [16]. Some studies focussed on the floristics and ecology of taxonomically diverse hosts at a single location, constructing a flora of Vestfold Hills [12] and King George's Island Potter Cove [2] respectively. Majewska et al. took a different approach by characterizing the epiphytic diatom ecology and flora of a small number of host taxa [3,13–15] across different locations [15].

In this study, diatom communities collected from diverse algal hosts belonging to different classes from two islands of the South Shetland archipelago, Deception (DI) and Livingston Islands (LI), were investigated. These islands differ strongly in their geology and geomorphology: DI is an active volcano, with comparatively young coastal ecosystems that undergo thermal disturbance due to volcanic activities on an irregular basis [17]. In contrast, LI harbours relatively undisturbed, pristine coastal locations with very steep inner slope moraines [18]. We attempt to interpret differences in epiphytic diatom compositions in a framework of environmental sorting effects resulting from differences in abiotic environments (island geology / coastline ecology, including depth), biotic interactions with macroalgal hosts (host phylogenetic position and/or gross morphology) and of a presumably low, but perhaps not negligible dispersal limitation in shaping these diatom communities. Although limited sampling in this distant region affects our study and limits the causal interpretability of statistical comparisons, just as it does for similar investigations in general, we attempt to disentangle the correlative contribution of these factors to community differences, while also substantially extending our diatom floristic knowledge of the Antarctic region.

## Materials and methods

A total of 36 macroalgal samples from 20 species and 2 macroscopically visible diatom community samples, i.e. diatom blooms attached to a substrate and visible to the naked eye, were taken in three consecutive annual expeditions (2017–2019) to the Antarctic South Shetland Islands, namely Deception (DI) and Livingston Island (LI) (Fig 1). The macroalgal samples were collected by hand from the intertidal range, as well as by snorkelling or by SCUBA divers at subtidal depths (Table 1). Simultaneously, temperature measurements were taken. All sampling permits (CPE-EIA-2013-7, CPE-EIA-2015-7, and CPE-EIA-2017-7) were issued by the Spanish Polar Committee within the Antarctic projects DISTANTCOM and BLUEBIO (CTM2013-42667/ANT and CTM2016-78901).

Macroalgae were obtained simultaneously with other benthic organisms at each sampling spot and pooled together in 1L receptacles, keeping different algal species separated from each other. At the wet lab, the specimens were sorted by *phylum* and identified to lowest taxon, usually species, following literature [19,20]. The species samples of one sampling site and day were

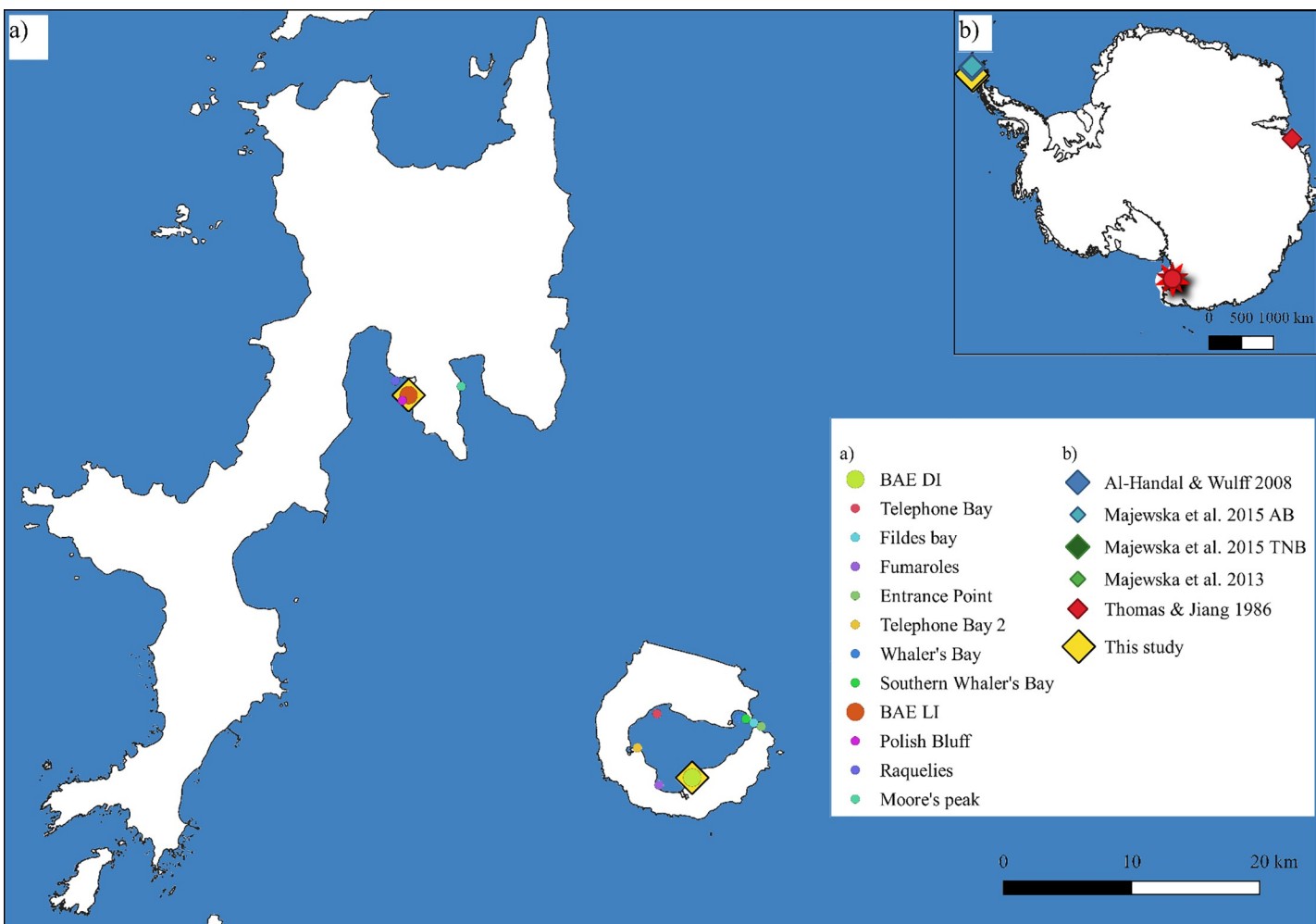

**Fig 1. Sampling localities.** a) Distribution of sampling sites in Livingston (LI) and Deception Islands (DI). b) sampling sites from previous studies from the literature. The yellow rectangle in b) shows the location of LI and DI. Map constructed with QGIS software.

kept in separate zip-style bags and frozen at -20˚C until further processing at the University of Barcelona. The macroalgal attributes of branching pattern, based on thallus morphology, and age, meaning the annuality of macroalgae (annual, biannual or pluriannual) of each species, were ascertained according to literature e.g. [19,21,22]. Epiphytic diatoms were extracted under laboratory conditions using a small part of the macroalgae, an overall appraisal of epiphyte incidence was made before scraping the surface into the receptacle with 80–100 ml of water depending on epiphyte density. The algae were also immersed and the samples were centrifuged [e.g. 23]. After this, the macroalgal part was extracted again for further use. Depending on epiphyte concentration, several aliquots were made and fixated using ethanol.

Diatoms were washed in distilled water to reduce remaining salinity. Samples were homogenized and centrifuged at 1000 rpm (Eppendorf centrifuge 5810 R, Eppendorf AG, Germany) for five minutes, and again filled up with deionized water. This procedure was repeated five times. Diatoms were prepared using the Friedrichs' [24] variation of Carr et al.'s method [25], using ten times diluted bleach (Domol Hygiene Reiniger, AGB Rossmann GmbH), based on 5% sodium hypochlorite as the undiluted oxidizing agent, with a treatment period of 30–45 min depending on the amount of organic matter present in the sample. The thus cleaned

**Table 1. Sample descriptions: Number of samples, replicates and macroalgal hosts.**

| | Host class | Host thallus morphology | Host annuality | Number of diatom taxa found (genera) | Number of samples (replicates) | Depth [m] | Year | Island |
|---|---|---|---|---|---|---|---|---|
| *Adenocystis utricularis* (Bory) Skottsberg) | Phaeo | Sac, Unb | A | 47 (22) | 1 (2) | 0 | 2018 | LI |
| *Ballia callitricha* (C. Agardh) Kützing | Rhodo | Fil, Bra | A, B | 41 (14) | 1 | 22.1 | 2018 | DI |
| *Berkeleya rutilans* (Trenthepohl ex Roth) Grrunow | Bacill. | Fil | A | 16 (7) | 1 | 4.5 | 2017 | DI |
| *Cystosphaera jacquinotii* (Montagne) Skottsberg | Rhodo | Lam, Bra | P | 25 (9) | 1 | 27 | 2017 | DI |
| *Delisea pulchra* (Greville) Montagne | Rhodo | Fil, Bra | P | 50 (16) | 4 | 21–23.4 | 2018, 2019 | LI |
| *Desmarestia anceps* Montagne | Phaeo | EBT | P | 44 (16) | 3 | 0–22 | 2017 | DI |
| *D. antarctica* R. L. Moe & P. C. Silva | Phaeo | EBT | P | 57 (24) | 2 | 0–13 | 2017, 2018 | DI, LI |
| *Desmarestia sp* J. V. Lamouroux | Phaeo | EBT | P | 18 (5) | 1 | 25 | 2017 | DI |
| *Gigartina skottsbergii* Setchell & N. L. Gardner | Rhodo | Lam, Bra | A,B | 53 (17) | 3 | 5.5–23.4 | 2017–19 | DI, LI |
| *Gymnogongrus cf. turquettii* Hariot | Rhodo | Lam, Bra | A, B | 47 (11) | 2 | 23–23.4 | 2018, 2019 | LI |
| *Himantothallus grandifolius* (A. Gepp & E.S. Gepp) Zinova | Phaeo | Lam, Unb | P | 50 (18) | 3 | 23–25 | 2017, 2018 | DI, LI |
| *Iridaea cordata* (Turner) Bory | Rhodo | Lam, Unb | A, B | 46 (15) | 3 | 0–25 | 2017, 2018 | DI, LI |
| *Monostroma hariotii* Gain | Chloro | Lam, Bra | A, B | 16 (7) | 1 | 23 | 2018 | LI |
| *Myriogramme manginii* (Gain) Skottsberg | Rhodo | Lam, Bra | Pp | 32 (11) | 1 | 22.1 | 2018 | LI |
| *Palmaria decipiens* (Reinsch) R. W. Ricker | Rhodo | Lam | Pp | 55 (23) | 3 | 2–17.5 | 2017, 2018 | DI, LI |
| *Picconiella plumosa* (Kylin) J. De Toni | Rhodo | BT | A, B | 40 (13) | 1 | 22.1 | 2018 | LI |
| *Plocamium cartilagineum* (Linnaeus) P.S. Dixon | Rhodo | EBT | A, B | 50 (15) | 4 | 22.1–25 | 2018, 2019 | LI |
| *P. cf. hookeri* Harvey | Rhodo | EBT | A, B | 23 (5) | 1 | 20 | 2018 | LI |
| *Pyropia endiviifolia* (A.Gepp & E.Gepp) H.G.Choi & M.S.Hwang | Rhodo | Lam | A, B | 39 (13) | 1 | 23 | 2018 | LI |
| *Brandinia mosimanniae* L.F. Fernandes & L. K. Procopiak [macroscopic] | Bacill. | Fil | A | 33 (17) | 1 | 8.2 | 2018 | DI |
| *In total* | | | | 131(85) | 38 | 0–25 | 2017–2019 | DI, LI |
| *In total (macroalgae)* | | | | 120 (47) | 36 | 0–25 | 2017–2019 | DI, LI |

Class names: Phaeo = Phaeophyceae, Rhodo = Rhodophyta, Chloro = Chlorophyta, Bacill = Bacillariophyceae. Sampling sites: LI = Livingston Island, DI = Deception Island. Morphological trends: EBT = Erect Branched Thallus, BT = Branched Thallus, Sac = Saccular Thallus, Fil = Filament, Lam = Laminar Thallus, Bra = Branched, Unb = Unbranched. Annual trends: P = Perennial, Pp = Pseudoperennial, B = Biannual, A = Annual.

diatom frustules were washed five times following the same procedure as before the bleach treatment. The frustule suspensions were then dripped onto coverslips, left to dry, quality checked and mounted using Norland Optical Adhesive 61 (refraction index = 1.56, Norland Products Inc., Cranbury, New Jersey, US).

Slide scanning methodology was modified after Kloster et al. [26] using a Metafer 4 slide scanner system (MetaSystems, Altlussheim, Germany) attached to an Axio Imager.Z2 micro-scope (Carl ZEISS AG, Oberkochen, Germany). The scans were made with a 63x objective

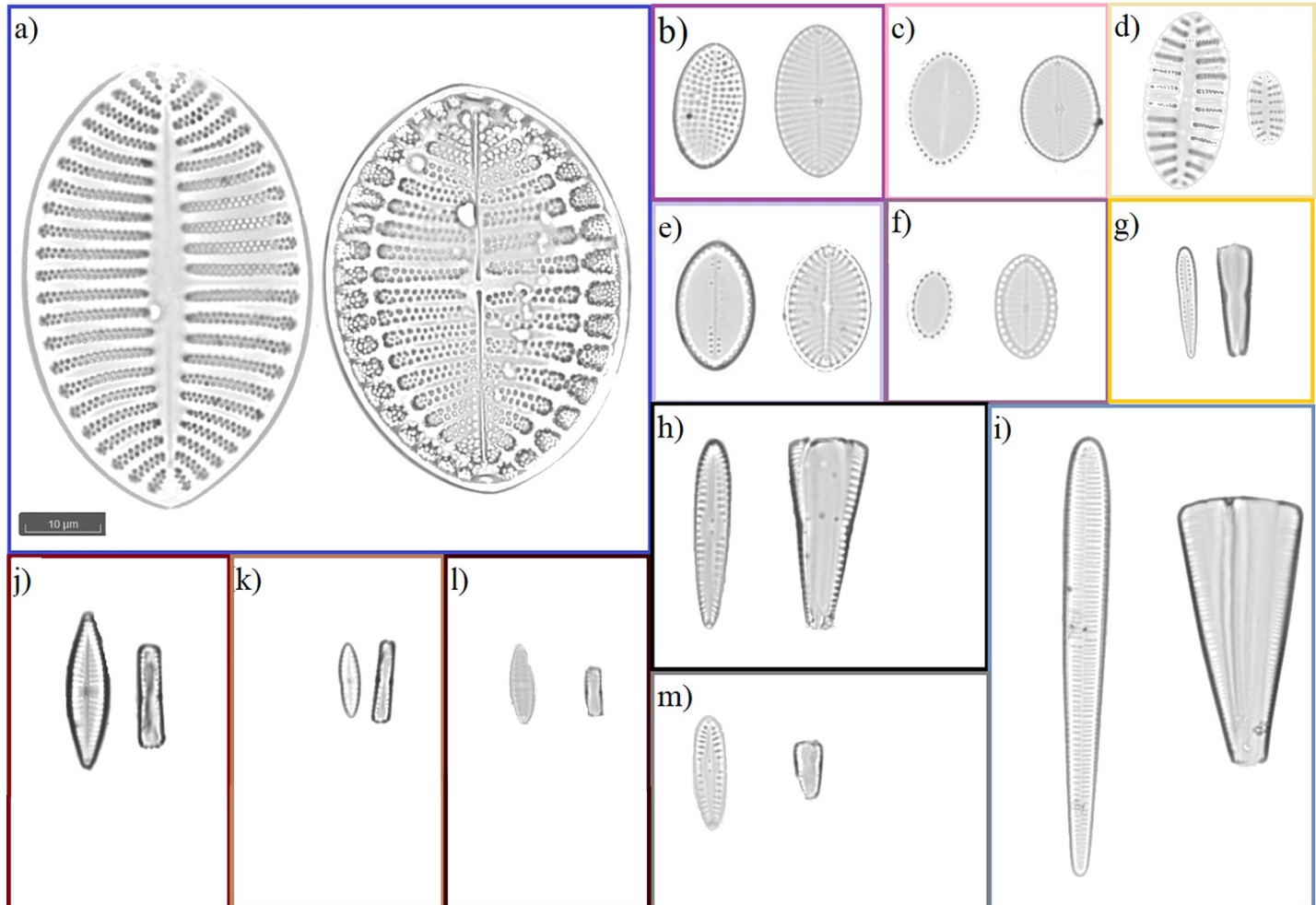

**Fig 2. Predominant diatoms found on Antarctic macrophytes.** Monoraphid diatoms shown in raphe and raphe-less valve view. Biraphids shown in valvar and pleural view. a) *Cocconeis fasciolata*, b) *Cocconeis californica var. californica*, c) *Cocconeis dallmannii*, d) *Cocconeis sp. 1*, e) *Cocconeis californica var. kerguelensis*, f) *Cocconeis melchioroides*, g) *Gomphonemopsis cf. ligowskii*, h) *Pseudogomphonema kamtschaticum*, i) *Licmophora gracilis*, j) *Navicula glaciei*, k) *Navicula incertata*, l) *Navicula perminuta*, m) *Pseudogomphonema sp. 1*.

(Zeiss Plan-Apochromat 1.4 with oil immersion) for an area of 54 x 75 visual fields, resulting in 4,050 images per slide, covering an area of 42.5 mm$^2$. For each field of view, images at 80 different focus levels were taken and combined to extended focus images. For the resulting images 980 pixels equal 100 µm, (e. g. see Fig 2 and images available in PANGAEA: https://doi.pangaea.de/10.1594/PANGAEA.925913.

The 4,050 extended focus depth images depicting each sample slide were stitched together to a virtual slide image using the Fiji image processing software [27]. First, the MIST plug-in [28–30] was used to calculate the exact relative position of the individual field of view-images. Subsequently, the tool MIST converter (Kloster, unpublished) was used to subdivide the slide into 3 segments of not more than 2 GB each and to process the position data for the last step, which utilized Grid / collection stitching ImageJ plug-in [31] for composing the virtual slide images. For each slide, this resulted in three virtual slide images which were then uploaded into the web-based annotation tool BIIGLE 2.0 [32].

In BIIGLE, the "random sampling" function was used to manually examine up to 400 randomly distributed sections of each virtual slide image at high magnification. Diatoms

contained within these sections were identified manually until ca. 500 identified specimens were reached for each sample, which mostly was the case during analysing the first of the three virtual slide image segments. In two instances material density on the slide was too low to allow for 500 annotations even after examining all three image segments comprising the sample, but the number of results was deemed sufficient to account for statistical significance. Diatoms were identified using epiphyte specific (e. g. [2,3,13–15]) and general [33,34] taxonomic bibliography to the lowest possible level. For each diatom specimen identified in this procedure, also their position (valvar vs. pleural view) and the presence of teratologic deformations was recorded. Teratologies refer to malformations, i.e. deviations from usual species-specific outline form or valve pattern, that can occur as a result of biotic and abiotic stresses [35].

Once all slides were identified, diatom inventories per virtual slide image were downloaded and, in cases where multiple images for the same slide were annotated, their counts were combined. The resulting inventories were turned into relative abundances (%). We calculated species richness and Shannon diversity [36]. To reduce influence of dominant taxa, relative abundances were square root transformed. Records of epiphytic diatom taxa were collated from previous studies undertaken around Antarctica (Fig 1), namely the South Shetland Islands (King George Island, in Admiralty Bay [15] and Potter Cove [2]), in McMurdo Sound (Terranova Bay [13–15] and Cape Evans [3]) and the Vestfold Hills (Davis Station [12]). Since the methodologies diverged in these studies and Thomas & Jiang did not provide numeric abundances, we used presence-absence data from the identified diatoms in each study. Slide scans and image cut-out of every single specimen identified in our study can be accessed in PANGAEA (https://doi.pangaea.de/10.1594/PANGAEA.925913). Statisical analyses, R Script and data matrices used are available in DRYAD (doi: https://doi.org/10.5061/dryad.ngf1vhhsm:).

## Statistical analysis

All statistical analyses were made with R software version 3.6.1 [37] on RStudio version 1.2.5019 [38]. Characterization by host species, thallus morphology, and branching pattern as well as annuality was made using IndVal algorithms [39]. The differential ternary graph showing species distributions of the epiphytic diatoms between three host classes (Phaeophyceae, Rhodophyta and Chlorophyta) was made using the 'ggtern' package [40]. Most multivariate analyses (non-metric dimensional scaling–nMDS-, principal component analysis—PCA) and analysis of similarity (ANOSIM), as well as the richness and diversity measures were calculated using the package 'vegan' [41]. Similarity of percentages (SIMPER) analyses were made using PRIMER software 7.0.13 (Primer-e Quest Research Limited, Auckland, New Zealand). Iterative hierarchical clustering was performed with 'cluster' [42] and 'pvclust' [43] packages. A Mantel test was performed combining the 'geosphere' [44] and 'vegan' packages When significant (P-value < 0.05), these values were further characterized as highly significant (***, $p < 0.001$), very significant (** $0.001 < p < 0.01$), or significant (*, $0.01 < p < 0.05$). The map in Fig 1 was constructed with QGIS software v. 3.16, [45] with the Quantarctica package (ADD_Coastline_res_line_Sliced) [46].

## Results

### Epiphytic diatom floristics and ecology

A total of 120 species of diatoms of 47 genera (S1 Table) were identified from 36 Antarctic macroalgae (Table 1). All macroalgae studied had varying degrees of epiphytic diatom colonization, with a range of 13 to 56 species per sample. The most frequent and predominant species of diatoms found in association with macroalgae (Fig 2) were generalist diatoms such as *Pseudogomphonema kamtschaticum* (Grunow) Medlin (up to 25% relative abundance in a

sample, present in all but one samples) or as yet undescribed species as *Navicula cf. perminuta* Grunow (up to 64% relative abundance in a sample, present in all samples) and *Pseudogomphonema* sp. 1 (up to 59% relative abundance in a sample, present in 29 samples). We recorded 19 diatom species not previously reported from these islands (S1 Table). Teratological frustules accounted for 0 to 2.3% of the counted cells. Diatoms had more teratologies on Rhodophyta (with an incidence of up to 2.4% of the sample and for 57.89% of all samples) than on Phaeophyceae (with an incidence of under 1%, in 32.89% of the samples).

Shannon diversity and diatom species richness (Table 2) did not follow a clear trend with location or depth. Neither host class nor host species was decisive for species richness. However, diatom species composition changed significantly for host class (Mantel statistic r = 0.45***). Shannon diversity on Phaeophyceae varied in a wider range (H' = 0.97–3.03) than on Rhodophyta (H' = 1.33–2.64), macroscopically visible diatoms (H' = 0.98–1.38) or Chlorophyta (H' = 1.49). Species richness was also more variable on Phaeophyceae (S = 13–56) than on Rhodophyta (S = 13–42), Bacillariophyceae (S = 16–33) and the Chlorophyta sample (S = 16). For some macroalgae species, different individual samples had similar diversity (such as *Delisea pulchra*, H' = 1.81–2.02) but varying species richness values (S = 19–28), or the other way around, as with *Iridaea cordata* (H' = 1.51–2.35, S = 18–23). In the case of *Himantothallus grandifolius*, diversity had a very wide range (H' = 0.97–2.66) as did species richness (H' = 13–39). All rarefaction curves calculated per host were saturated (S1 Fig) and Rhodophyta had the highest species richness of all macrophytes studied.

The ternary plot illustrates the preferences between the host groups for the predominant species (Fig 3), where only *Pseudogomphonema kamtschaticum* showed no host preference at all (Fig 3). On the other hand, some diatom-macroalgae class relationships are rather specific, as *Licmophora gracilis* was found mostly on the Chlorophyta, *Cocconeis melchioroides* on Rhodophyta, and *Cocconeis fasciolata* on Phaeophyceae. The ternary plot (Fig 3) further shows that most diatom taxa were shared amongst Phaeophyceae and Rhodophyta. Since only one Chlorophyta was sampled, some or all of these might represent host generalist taxa which would also be found on Chlorophyta with more sampling effort.

ANOSIM showed that host class had the highest impact of the macroalgal characteristics on diatom distribution (R = 0.47***). Host branching patterns (R = 0.17*) and annuality (R = 0.23***) also affected the diatom community to varying degrees. As the rarefaction curves (S1 Fig) show, only Phaeophyceae and Rhodophyta arrived at saturation levels with the samples explored. When comparing the diatom communities on these macroalgal classes (Phaeophyceae and Rhodophyta), only Rhodophyta had significant ANOSIM values, e.g. variation inside the class and between species (Table 3). Diatom communities growing on Rhodophyta were found to be substantially shaped by locality (R = 0.39***), year (R = 0.38***), and annuality of the host (R = 0.25***).

The nMDS multivariate analysis (Fig 4) performed on diatom communities showed a small degree of differentiation depending on macroalgal host. A two-dimensional solution was sufficient due to the low stress value recorded (0.16). On the other hand, a SIMPER analysis on predominant diatoms showed very high standard deviation levels (S2 Table). When looking into the most abundantly sampled Rhodophyta and Phaeophyceae (Table 4), the SIMPER analysis results showed that *Navicula perminuta*, *Gomphonemopsis ligowskii* and *Cocconeis melchioroides* were the most significant contributors to the average dissimilarity.

## Diatom distribution in Antarctica

**Diatoms in the South Shetland Islands.** The annual mean temperature of both islands was 2˚C, but the temperature range in DI comprised 0–4˚C (increasing even more towards the

**Table 2. Sampling site characterization of depth and temperature (T) and diatom epiphyte richness (S) and diversity (H') found on each macroalgal host.**

| Locations | Depth (m) | T (°C) | Macroalgal host | Species diversity (H') | S |
|---|---|---|---|---|---|
| Livingston–Raquelies | 13 | / | *Adenocystis utricularis* | 1.76 | 45 |
| Deception–Antarctic base | 4.5 | 3 | *Berkeleya rutilans* | 0.98 | 16 |
| Deception–Fumaroles | 8.2 | * | *Brandinia sp.* | 1.38 | 33 |
| Livingston–Moore's peak | 22.1 | 2 | *Ballia callitricha* | 2.58 | 40 |
| Deception–Whaler's bay | 25 | 2 | *Cystosphaera jacquinottii* | 1.74 | 24 |
| Livingston–Polish Bluff | 21 | 3 | *Delisea pulchra* | 2.02 | 27 |
| Livingston–Polish Bluff | 22.1 | 1 | *Delisea pulchra* | 1.81 | 19 |
| Livingston–Moore's peak | 23 | 2 | *Delisea pulchra* | 1.93 | 20 |
| Livingston–Polish Bluff | 23.4 | 2 | *Delisea pulchra* | 2.02 | 28 |
| Deception–front of base | 0 | / | *Desmarestia anceps* | 2.17 | 30 |
| Deception–front of base | 0 | / | *Desmarestia anceps* | 1.51 | 13 |
| Deception–Fildes bay | 22 | 3 | *Desmarestia anceps* | 1.33 | 22 |
| Livingston–Raquelies | 13 | 2 | *Desmarestia antactica* | 3.03 | 56 |
| Deception–Antarctic base | 0 | / | *Desmarestia antarctica* | 1.76 | 17 |
| Deception–Whaler's bay | 25 | 2 | *Desmarestia sp* | 2.08 | 17 |
| Deception–Seal colony | 5.5 | 4 | *Gigartina skottsbergii* | 2.56 | 42 |
| Livingston–Moore's peak | 22.1 | 2 | *Gigartina skottsbergii* | 2.18 | 21 |
| Livingston–Polish Bluff | 23.4 | 2 | *Gigartina skottsbergii* | 1.34 | 13 |
| Livingston–Polish Bluff | 23.4 | 2 | *Gymnogogrus turquettii* | 1.77 | 32 |
| Livingston–Moore's peak | 23 | 2 | *Gymnogongrus turquettii* | 2.09 | 37 |
| Deception–Fildes bay | 25 | 2 | *Himantothallus grandifolius* | 0.97 | 13 |
| Deception–Whaler's bay | 25 | 2 | *Himantothallus grandifolius* | 2.66 | 39 |
| Livingston–Moore's peak | 23 | 2 | *Himantothallus grandifolius* | 2.05 | 23 |
| Deception–front of base | 0 | / | *Iridaea cordata* | 1.51 | 18 |
| Deception–Whaler's bay | 25 | 2 | *Iridaea cordata* | 2.35 | 23 |
| Livingston–Moore's peak | 22.1 | 1 | *Iridaea cordata* | 2.25 | 23 |
| Livingston–Moore's peak | 23 | 2 | *Monostroma hariotii* | 1.49 | 16 |
| Livingston–Moore's peak | 22.1 | 2 | *Myriogramme cf. manguinii* | 2.36 | 31 |
| Deception–Drum | 17.5 | 2 | *Palmaria decipiens* | 1.96 | 30 |
| Deception–Telephone bay | 14.1 | 3 | *Palmaria decipiens* | 1.43 | 30 |
| Livingston–Antarctic Base | 2 | 2 | *Palmaria decipiens* | 1.85 | 26 |
| Livingston–Moore's peak | 22.1 | 2 | *Piccionella plumosa* | 2.49 | 39 |
| Livingston–Raquelies | 25 | 2 | *Plocamium cartilagineum* | 2.18 | 22 |
| Livingston–Moore's peak | 22.1 | 1 | *Plocamium cartilagineum* | 2.19 | 30 |
| Livingston–Moore's peak | 23 | 2 | *Plocamium cartilagineum* | 2.31 | 30 |
| Livingston–Polish Bluff | 23.4 | 2 | *Plocamium cartilagineum* | 1.85 | 25 |
| Livingston–Moore's peak | 20 | 1 | *Plocamium cf. hookerii* | 2.01 | 23 |
| Livingston–Moore's peak | 23 | 2 | *Pyropia endiviifolia* | 2.64 | 38 |

Intertidal temperature was not recorded [/].

* Taken in the fumaroles, temperature estimated between 40–80°C.

fumaroles, but not recorded), and 1–3°C in LI. A total of 15 samples came from DI (diatom epiphyte taxa n = 94) and 23 samples from LI (diatom epiphyte taxa n = 82), and 66 diatom taxa (21 genera) were shared between both islands. Diatom compositions of Rhodophyta and Phaeophyceae from DI and LI clustered together, separated from the Chlorophyta and macroscopic Bacillariophyceae samples, in hierarchical cluster analysis (distance = Euclidean,

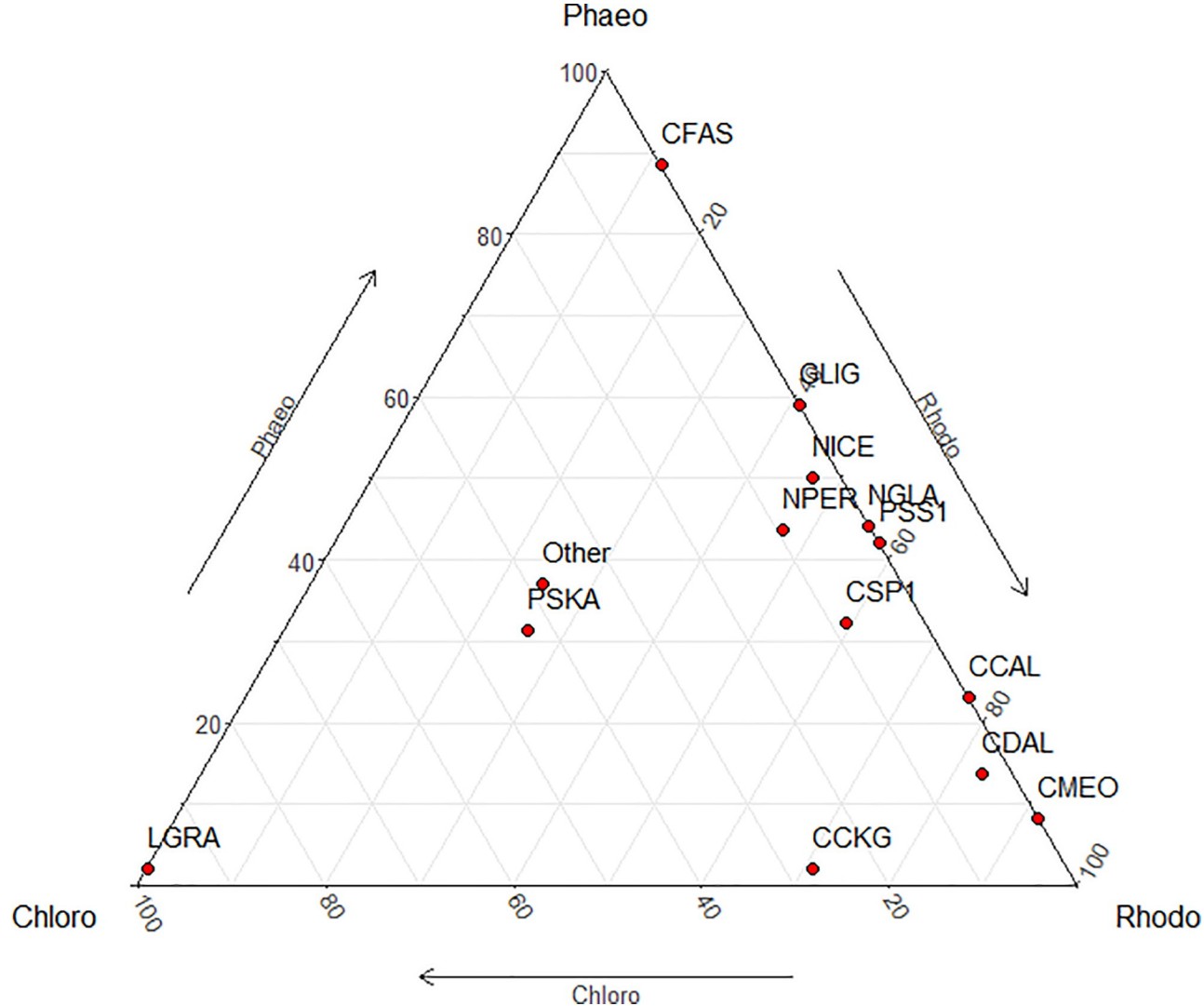

**Fig 3. Ternary plot of predominant epiphytic diatoms shared between Rhodophyta (Rhodo), Phaeophyceaes (Phaeo) and Chlorophyta (Chloro).** Species codes: CCAL = *Cocconeis californica*, CCKG = *Cocconeis californica var. kerguelensis*, CDAL = *Cocconeis fasciolata*, CFAS = *Cocconeis dallmanii*, CMEO = *Cocconeis melchioroides*, CSP1 = *Cocconeis sp. 1*, GLIG = *Gomphonemopsis ligowskii*, LGRA = *Licmophora gracilis*, NGLA = *Navicula glacialis*, NICE = *Navicula incertata*, NPER = *Navicula perminuta*, PSKA = *Pseudogomphonema kamtschaticum*, PSS1 = *Pseudogomphonema sp 1*, Other = diatom species in under 2% frequency and abundance.

cluster = average linkage, bootstrap = 94%, not shown). A redundancy analysis (RDA), on the other hand, showed a separation between DI and LI communities (Fig 5). This was corroborated by the results of the Mantel test (geographical distance matrix vs. diatom communities,

**Table 3. ANOSIM test results performed on communities from Phaeophyceae (Phae) and Rhodophyta (Rhod) hosts.**

|  | Loc | | Depth | | Depth Int | | Year | | Host Morph. | | Host Branch | | Host Annuality | |
|---|---|---|---|---|---|---|---|---|---|---|---|---|---|---|
|  | Phae | Rhod | Phae | Rhod | Phae | Rhod | Phae | Rhod | Phae | Rhod | Phae | Rhod | Phae | Rhod |
| **R** | 0.03 | **0.39** | 0.19 | **0.26** | 0.12 | **0.36** | 0.03 | **0.38** | -0.02 | 0.12 | -0.05 | **0.29** | 0.09 | **0.37** |
| **p-val** | >0.05 | 0.002 | >0.05 | 0.03 | >0.05 | 0.01 | >0.05 | 0.0008 | >0.05 | >0.05 | >0.05 | 0.007 | >0.05 | 0.003 |

Loc = Location, Depth Int = Depth Interval, Host Morph = Host Morphology, Host Branch = Host Branching pattern, Host Annual = Host Annuality. Significant values are highlighted in bold.

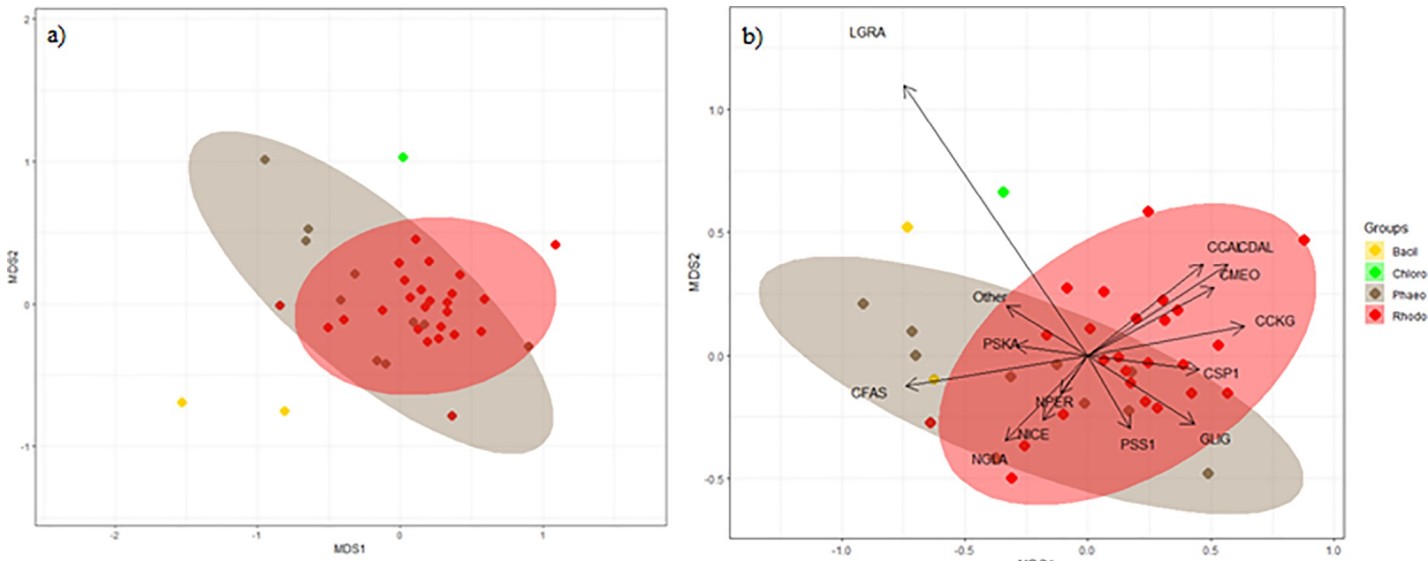

**Fig 4. nMDS of the diatom communities.** a) complete set and b) predominant diatoms (square root transformed). Bacill = macroscopically visible Bacillariophyceae, Chloro = Chlorophyta, Phaeo = Phaeophyceae, Rhodo = Rhodophyta. Species codes: CCAL = *Cocconeis californica*, CCKG = *Cocconeis californica var. kerguelensis*, CDAL = *Cocconeis dallmanii*, CFAS = *Cocconeis fasciolata*, CMEO = *Cocconeis melchioroides*, CSP1 = *Cocconeis sp. 1*, GLIG = *Gomphonemopsis ligowskii*, LGRA = *Licmophora gracilis*, NGLA = *Navicula glacialis*, NICE = *Navicula incertata*, NPER = *Navicula perminuta*, PSKA = *Pseudogomphonema kamtschaticum*, PSS1 = *Pseudogomphonema sp 1*, Other = diatom species in under 2% frequency and abundance.

$r = 0.299^{***}$). The frequency of teratologies found was higher in LI (56.52% of samples had teratological cells, arriving at 2.3% of incidence in a sample) than in DI (66.67% of samples had teratological cells, with an incidence between 0–1% of the samples). However, only samples from Deception island did not arrive to 500 valves due to sparse epiphyte concentration.

Given that we had not enough specimens of macroscopic diatom colonies (n = 2) and Chlorophyta (n = 1), only Rhodophyta and Phaeophyceae samples were considered henceforth. When comparing the depth distribution of predominant diatom taxa (Fig 6), frequent or abundant diatom taxa, or both, differences between the samples from LI and DI became apparent. Larger diatoms, such as *Cocconeis fasciolata* (Ehrenberg) N. E. Brown or *C. antiqua*

**Table 4. Average abundance and dissimilarity of diatom communities from Rhodophyta (Rhod) and Phaeophyceae (Phae).**

| Taxa | Average abundance | | Average dissimilarity | SD | Contribution (%) | Cumulated (%) |
|---|---|---|---|---|---|---|
| | **Phaes** | **Rhod** | | | | |
| *Navicula perminuta* | 18.36 | 15.88 | 8.48 | 1.05 | 11.65 | 11.65 |
| *Gomphonemopsis ligowskii* | 12.19 | 8.79 | 7.42 | 0.94 | 10.19 | 21.84 |
| *Cocconeis melchioroides* | 1.44 | 13.65 | 6.48 | 0.88 | 8.90 | 30.73 |
| *Pseudogomphonema* sp. 1 | 10.72 | 12.21 | 6.44 | 1.05 | 8.84 | 39.58 |
| *Cocconeis fasciolata* | 10.28 | 0.94 | 5.13 | 0.66 | 7.04 | 46.61 |
| *Pseudogomphonema kamtschaticum* | 8.80 | 4.77 | 4.01 | 1.09 | 5.51 | 52.12 |
| *Cocconeis californica var. kerguelensis* | 0.23 | 7.38 | 3.68 | 0.56 | 5.05 | 57.17 |
| *Tabularia tabulata* | 6.80 | 0.09 | 3.43 | 0.32 | 4.71 | 61.88 |
| *Cocconeis dallmannii* | 0.92 | 5.20 | 2.69 | 0.62 | 3.69 | 65.57 |
| *Cocconeis costata* | 4.94 | 2.01 | 2.53 | 0.73 | 3.48 | 69.05 |
| *Cocconeis californica* | 1.21 | 4.13 | 2.32 | 0.47 | 3.19 | 72.23 |

Species ordered in decreasing and cumulated contributions (SIMPER analysis). SD = Standard deviation.

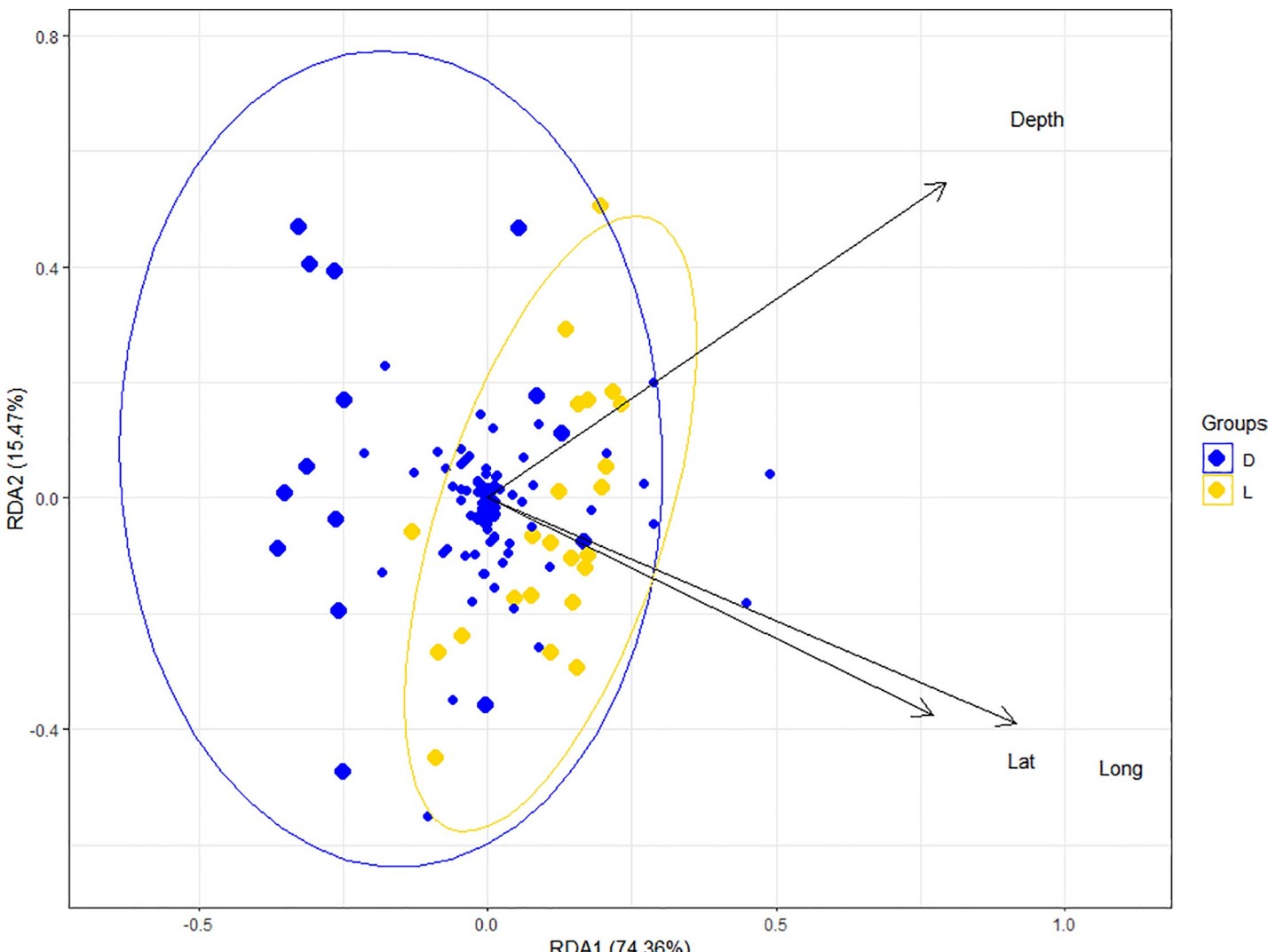

**Fig 5. RDA with the first two axes explaining 89.83% of total variance Eigen values of axis 1 = 2.80 and axis 2 = 0.582.** DI = Deception island, LI = Livingston island.

Tempère & Brun, were found on samples located in shallower locations in DI, while smaller diatoms, as *Navicula cf. perminuta* Grunow were mostly found in comparatively deeper samples. On the contrary, in LI, this depth-cell size trend was reversed. The differentiation of diatom communities with respect to sampling depth was significant as well (Mantel statistic r = 0.260***).

ANOSIM showed that the importance of factors determining diatom community composition differed between both islands (Table 5). The predominant factor was host algae species on LI (R = 0.7 ***) and depth on DI (R = 0.54***). Host class was significant in both locations (DI R = 0.38 ***, LI R = 0.4 *). SIMPER analysis (Table 6) further showed an average dissimilarity of 70.71% between islands and an intra-island dissimilarity of 59.51% (LI) and 74.80% (DI). The most characteristic diatom species for DI were *Cocconeis melchioroides*, *Pseudogomphonema* sp. 1 and *Gomphonemopsis ligowskii*. In LI, the predominant diatom was *Navicula perminuta*. Both islands had saturated rarefaction curves (S2 Fig), and DI seemed to have the richest diatom community.

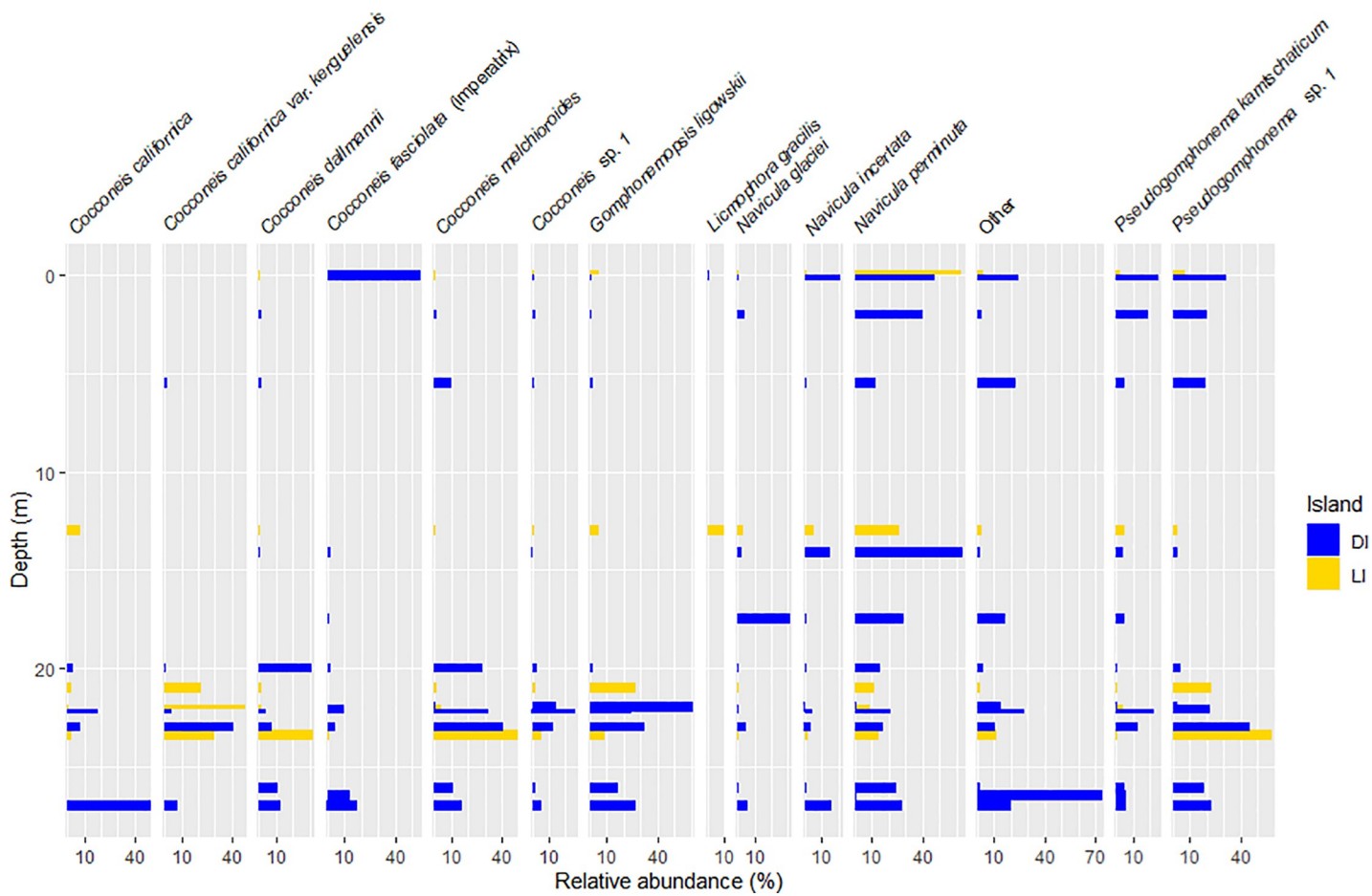

**Fig 6. Depth diagram of diatom distribution in Deception and Livingston Island.** The rest of diatoms found are summarized in the "Other" panel.

**Antarctic epiphytic diatoms in the literature.** A Mantel test on a presence-absence database created from diatom community composition depending on study location showed significance in the richness of the epiphytic diatom communities according to geographical GPS spherical trigonometric distance (Mantel statistic r = 0.4675**). When comparing the diatom composition of the samples, the sample of Vestfold Hills was the most diversified from the rest when comparing epiphytic diatom composition using a hierarchical clustering (Fig 7, S3–S5 Tables). The samples from McMurdo Sound (MS) clustered together, and our samples clustered with the South Shetland Islands (SSI) sample from Potter Cove. The sample from Admiralty Bay, however, clustered with the diatom composition found on macroalgae from MS.

**Table 5. ANOSIM test results performed on communities from Deception (DI) and Livingston Island (LI).**

|         | Host algae | | Host class | | Depth [m] | | Year | | Host morph. | | Host branch | | Host annual | |
|---------|------|------|------|------|------|------|------|------|------|------|------|------|------|------|
|         | D    | L    | D    | L    | D    | L    | D    | L    | D    | L    | D    | L    | D    | L    |
| **R**       | 0.04 | **0.70** | **0.38** | **0.40** | **0.54** | 0.12 | 0.26 | 0.22 | 0.21 | 0.13 | 0.03 | 0.04 | 0.15 | 0.13 |
| **p-value** | >0.05 | 0.0001 | 0.001 | 0.01 | 0.008 | >0.05 | >0.05 | >0.05 | >0.05 | >0.05 | >0.05 | >0.05 | >0.05 | >0.05 |

Significant values are highlighted in bold.

**Table 6. Breakdown of average dissimilarity between epiphytic diatoms in Deception and Livingston Island locations (SIMPER).**

| Taxa | Average abundance | | Average dissimilarity | SD | Contribution (%) | Cumulated (%) |
|---|---|---|---|---|---|---|
| | Deception | Livingston | | | | |
| *Navicula perminuta* | 16.89 | 14.90 | 8.00 | 1.01 | 10.44 | 10.44 |
| *Gomphonemopsis ligowskii* | 6.11 | 10.91 | 6.94 | 0.98 | 9.05 | 19.50 |
| *Pseudogomphonema* sp.1 | 8.08 | 12.51 | 6.39 | 0.99 | 8.34 | 27.83 |
| *Cocconeis melchioroides* | 2.71 | 13.09 | 6.37 | 0.87 | 8.31 | 36.14 |
| *Cocconeis fasciolata* | 7.64 | 0.97 | 3.91 | 0.56 | 5.11 | 41.25 |
| *Cocconeis californica var. kerguelensis* | 0.83 | 7.36 | 3.78 | 0.57 | 4.94 | 46.19 |
| *Pseudogomphonema kamtschaticum* | 6.19 | 5.54 | 3.42 | 1.00 | 4.46 | 50.65 |
| *Cocconeis californica* | 3.64 | 2.48 | 2.79 | 0.47 | 3.64 | 54.29 |
| *Cocconeis dallmannii* | 0.96 | 5.22 | 2.79 | 0.63 | 3.64 | 57.92 |
| *Berkeleya rutilans* | 5.18 | 0.00 | 2.60 | 0.27 | 3.39 | 61.31 |
| *Tabularia tabulata* | 5.00 | 0.09 | 2.54 | 0.27 | 3.31 | 64.62 |
| *Brandinia* | 4.96 | 0.01 | 2.49 | 0.28 | 3.25 | 67.87 |
| *Navicula incertata* | 4.49 | 1.78 | 2.24 | 0.73 | 2.92 | 70.79 |

Species ordered in decreasing and cumulated contributions (SIMPER analysis). SD = standard deviation.

ANOSIM showed significant differences in diatom composition by study after controlling for geographic effect (R = 0.78***), after controlling for geographic effects study made out (R = 0.81***). For further characterization, a SIMPER analysis was used following the distribution around Antarctica. Samples from SSI, MS, and Vestfold Hills (VH) showed significant differences among each other, with MS and VH having the highest average dissimilarity (99.51), followed by South Shetland Islands and VH (98.89), and SSI and MS being the lowest (80.54). MS was characterised by the most frequent taxa *Fragilariopsis nana*, *Cocconeis fasciolata* and *Pseudogomphonema kamtschaticum*. In VH only *Nitzschia lecointei* seemed to be characteristic. The most frequent taxa in SSI, which includes the diatoms of our study, were *Navicula perminuta* and *Cocconeis melchioroides*. The comparison of diversity (S6 Table) showed that the DI samples had the highest species richness overall (S = 94) and a relatively high Shannon diversity (H' = 3.16) compared to the other SSI samples (H' = 2.63–2.90). SSI and MS sites had similar values of diversity (H' = 2.63–3.87) and richness (S = 45–118).

## Discussion

The total number of taxa identified in this study, 129 species and 44 genera, exceeds the number of taxa in previous Antarctic epiphytic diatom studies. Even after eliminating the diatom samples from the dataset, a total of 120 species and 42 genera of epiphytic diatoms were identified on macroalgal samples, still surpassing the diversity found in previous studies. This could be an effect of a broader sampling along the depth gradient, of a high richness of macroalgal species investigated in this study, or the gentle preparation method used. A partial explanation of high diatom species richness in Antarctic-Subantarctic marine benthos might be the unusually high nutrient concentrations (especially of nitrate) surrounding the Antarctic peninsula [22] in combination with higher iron levels [47]. The high richness of macroalgal species investigated in this study in combination with the ecological niche diversity is, however, probably more important. Majewska et al [3,15] studies were only based on three Rhodophyta taxa. The study on epiphytic diatoms in Vestfold Hills [12] had 17 host species, but epiphytic diatom species numbers remained low as the authors only reported diatoms commonly found on different types of macroalgae and sea ice. The most comparable study would be Al-Handall et al.

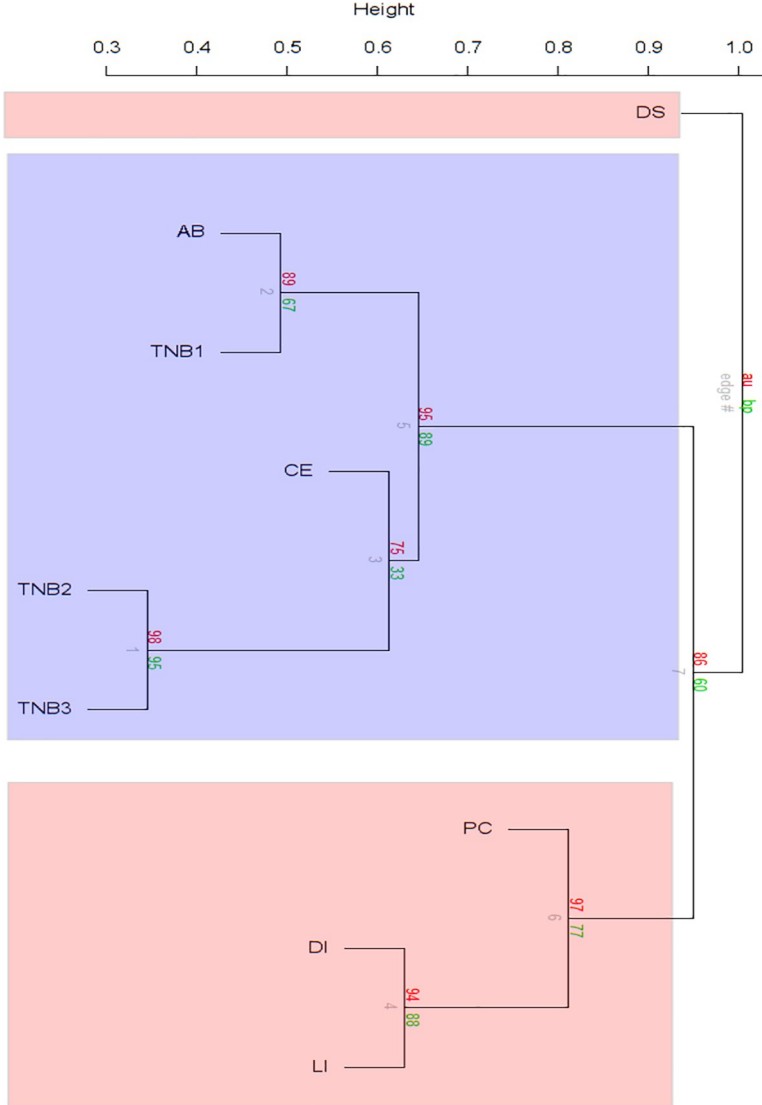

**Fig 7. Hierarchical clustering calculated with 10,000 permutations with the studies of Davis Station (DS), Terra Nova Bay (TNB1, 2, 3), and South Shetland Islands (Deception [DI], Livingston [LI], and King George Island, divided into Potter Cove [PC] and Admiralty Bay [AB]) with presence-absence data aggregated at the study level (n total = 192).**

[2] (19 host species and individual samples), which listed 50 species, compared to our total of 20 macrophyte taxa (and 38 samples).

In spite of previous studies, 20 diatom species were recorded for the first time in DI and LI (S1 Table, bold). Most of them pertained to the *Cocconeis* Ehrenberg genus, a monoraphid and mostly epiphytic diatom [48]. This genus was also predominant in previous studies [3,13–15]. One frequent diatom taxon was identified as an unknown species. *Pseudogomphonema* sp. 1 was smaller than *Pseudogomphonema plinskii* Witkowski, Metzelin & Lange-Bertalot and the endophytic diatom found inside the macroalgal genus *Neoabbottiella* [49] and could be yet undescribed. In contrast with the epiphytic diatom studies, usual proxies for sea-ice as *Fragilariopsis curta* [50] and *Thalassiosira antarctica* [51] were not found in as much predominance as, for instance, in Majewska et al. [3].

## Epiphytic diatom floristics and ecology

Seaweeds respond to changes in several ways, including by secreting secondary metabolites with antibiotic or antifouling activities on surfaces susceptible to epiphytic invasion [19–21,52]. This could activate the acclimatisation mechanisms of epiphytic diatoms and co-specialization could be prompted. In our study however, the difference between both islands could reflect different taxonomic coverages of sampling, rather than genuine biogeographic signal: whereas most of the samples from LI were Rhodophyta (>82%), in DI the proportions were 40% Rhodophyta and 46% Phaeophyceae. Therefore, the DI samples contained more information on the effect of host at the higher taxonomic level and could be better compared with Al-Handal and Wulff [2], while LI samples more information at lower taxonomic level within red algae, showing more similarities to the Majewska results [3].

Apart from taxonomic identity, also branching pattern and annuality of the host have previously been found to shape epiphyte communities [53]. We found such associations, too, like some species of *Cocconeis* occurring only on branched Rhodophyta hosts (e.g. *Desmarestia* or *Plocamium*), which coincides with findings in other Antarctic and worldwide marine epiphytic diatom studies [3,54–57]. In our study, the effect of branching pattern and annuality was only significant on Rhodophyta and not on Phaeophyceae hosts. As the host age increases, so does the colonization by a mature biofilm community [58]. Maturity of the biofilm, and thus organism position inside a polysaccharide casing, could also protect the community living in it, as shown in several heavy metal studies [59,60]. This might partially explain the annuality effect.

It is instrumental to compare study designs across Antarctic-Subantarctic epiphyte diatom studies. Whereas most sampling campaigns in these distant regions are opportunistic by necessity [2,12], Majewska and collaborators [3,13–15] deliberately focused on three macroalgal host species, systematically capturing epiphyte variability on these selected hosts. In contrast, the present study sampled non-selectively, but the so far broadest diversity of host taxa, and at least some of them repeatedly. Comparing results from both types of approaches, it becomes clear that a systematic and repeated sampling of a broader range of host species will be required for a final clarification of the specificity of host-epiphyte associations. With respect to host-trait effects on epiphyte communities, it would be interesting to more systematically compare branched vs unbranched red and brown seaweed host taxa (for instance the Rhodophyta *Plocamium cartilagineum* vs. *Iridaea chordata*, and the Phaeophyceae *Desmarestia antarctica* vs. *Hymantothallus grandifolius*).

## Geographical distribution of diatoms in Antarctica

The total species richness found in DI (93) far exceeded the expectations for an extreme environment, being lower in LI (82). In LI a higher variability in light impact could have been expected because of glacier inputs [61]. Better micronutrient supply due to the volcanic exudations on DI might have increased the number of species [47]. On the other hand, substrate consistency could also have an effect on diatom colonization. The fine (lapilli) consistency in DI causes quickly changing light intensities, since the substrate can quickly redeposit itself after being moved. In contrast, the light influx of LI does not depend on movement of lapilli, but varies due to the input from time constrained glacier melt [62], thus providing more stable irradiation for macroalgae and epiphytic diatoms during the sampled Antarctic summer.

Deception island (DI) is a quiescent volcano, with a semi-submerged cone. The caldera is only 180 m deep [63] and has active hydrothermal vent activity [64]. This has been found to increase the bioavailability of trace elements [65], and also to increase colloid suspension because of the fine sediment or lapilli [64]. The existence of active fumaroles also increases the temperature range measured in the water and substrate [66], thus further segregating potential

ecological niches in the ecosystem. Another important difference between the two islands is that LI has a slightly higher tide amplitude and narrower range of water temperature than Deception island [67].

The difference in diatom taxa of both islands was smaller than expected and also smaller than the host effect. Host species that were sampled in DI and LI once or more were compared between and within locations (*Desmarestia antarctica*, *Gigartina skottsbergii*, *Hymantothallus grandifolius*, *Iridaea cordata* and *Palmaria decipiens*) and showed that the dissimilarity among DI samples was greater than in LI or in comparison between DI and LI. This might point to the environmental variability inside the DI caldera creating more ecological niches for diatom species to fill [68], but unfortunately, physicochemical and light intensity measurements which would be needed to substantiate this are not available.

On a broader geographic scale, similarity of diatom communities around Antarctica was strongly dependent on study. Using presently available data, it is not possible to separate geographic differences from environmental effects and possible effects of methodological differences among studies (further discussed in Closing methodological remarks). It is of course to be seen as a tentative comparison of epiphytic diatom distribution around Antarctica, since other variables concerning seasonality and physicochemical composition of the waters in each of the studies was mostly unavailable and further, synchronous studies should be made to answer the question of epiphytic diatom biogeography around Antarctica. This study would be a first approach, but as discussed, new and more standardized / synchronised efforts should be made in the future to obtain a clear picture on the ecological variations of epiphytic diatoms along the Antarctic coastline.

## Closing methodological remarks

One of the most striking observations of our study was the strong effect of study upon epiphytic diatom communities. As discussed above, the exact cause of this study effect is difficult to pin down based on presently available data, but preparation method might be part of it. The dehydration method used by Majewska and collaborators [3,5,13–15] permitted a quantitative in situ observation but could potentially lead to overlooking taxa growing in lower layers of the established biofilm on the host algae. As previously discussed by Majewska et al [3], the reduced diatom species richness in Al-Handal & Wulff [2] and Thomas & Jiang [12] could be an effect of dissolution of lightly silicified frustules. Although silicate is known to dissolve faster in alkaline than in acidic milieu, Carr [25] and Friedrichs [24] found that a short-time bleach treatment, as used in this study, is more gentle to diatom frustules than commonly applied harsh oxidizing acid treatments which was used by previous ones. Parallel preparations from the same sample using both types of approaches in the future would be useful to test whether the effect of preparation treatment is indeed the dominant cause of study effect. Once this has been clarified, a clearer recommendation for standardizing the methodology of epiphytic diatom preparation can be given which will be important to improve the comparability of results among different studies.

Another methodological difference of our study from previous ones was the use of virtual slide microscopy and web-based manual taxonomic annotation. We did not systematically test this effect in these studies, but checking individual samples both on the light microscope and in the slide scans indicates that this is not causing a major bias for observing taxonomic composition (a study systematically comparing this effect is presently in preparation). We think that this methodology has some potential advantages for the future. For instance, a digital image of every single frustule identified in this study is available in PANGAEA (doi: https://doi.pangaea.de/10.1594/PANGAEA.925913). Future studies making literature comparisons,

like attempted also above, will thus not only have presence-absence records, but also every one of these images, making it even possible to re-identify any or all frustules as deemed necessary. This can, in the long run, when such data sets accumulate, contribute a lot to transparency and comparability among different studies.

In conclusion, in this study we compared epiphytic diatom communities living on several macroalgae in Deception and Livingston Island. We found that the number of species in DI samples exceeded those from LI and from previous studies. The former observation may point to a higher proportion of niches found on the volcanic island. The second one would be explained by a gentler preparation method, though this needs a clear causal confirmation in the future.

## Supporting information

**S1 Fig. Rarefaction curves of macrophytes hosts.** Rhodo = Rhodophyta, phaeo = Phaeophyceae, bacil = Bacillariophyceae, chloro = Chlorophyta. (TIF)

**S2 Fig. Rarefaction curves of sample location in this study.** Dec = Deception island, Liv = Livingston island. (TIF)

**S1 Table. Epiphytic diatom composition and frequency on Antarctic macroalgae, specific proportion and occurrence on Deception (DI) and Livingston Island (LI).** Pheaophyceae (Phaeo, n = 10), Rhodophyta (Rhodo, n = 25), Bacillariophyceae (Bac, n = 2) and Chlorophytas (Chlo, n = 1) have been investigated. Macroalgal host: Au = *Adenocystis utricularis*, Bc = *Ballia callitricha*, Cj = *Cystosphaera jacquinotii*, Dp = *Delisea pulchra*, Dan = *Desmarestia anceps*, Dant = *Desmarestia antarctica*, Ds = *Desmarestia sp.*, Gs = *Gigartina skottsbergii*, Gt = *Gymnogongrus turquettii*, Hg = *Himantothallus grandifolius*, Ic = *Iridaea cordata*, Mh = *Monostroma hariotii*, Mm = *Myriogramme cf. manginii*, Pd = *Palmaria decipiens*, Pp = *Piccionella plumosa*, Pc = *Plocamium cartilagineum*, Ph = *Plocamium cf. hookeri*, Pe = *Pyropia endiviifolia*. Bold text shows first records in DI and LI. (DOCX)

**S2 Table. SIMPER analysis of comparison of predominant diatom species in LI and DI.** (DOCX)

**S3 Table. SIMPER analysis of comparison of epiphytic diatom communities in the South Shetland Islands (SSI, n = 4) and Vestfold Hills (VH, n = 1).** (DOCX)

**S4 Table. SIMPER analysis of comparison of diatom communities in the South Shetland Islands (SSI, n = 4) and MacMurdo Sound (MMS, n = 4).** (DOCX)

**S5 Table. SIMPER analysis of comparison of epiphytic diatom communities in the Vestfold Hills (VH, n = 1) and MacMurdo Sound (MMS, n = 4).** (DOCX)

**S6 Table. Diversity/Entropy indices of the compared studies.** * = contains only macroalgal information, not Bacillariophyceae. | = total data of the study. (DOCX)

**S7 Table. Mantel result comparison of parameters and diatom abundance.** (DOCX)

## Acknowledgments

We would like to thank all the scientists involved in sampling Antarctic macroalgae, particularly Elisenda Ballesté and Blanca Figuerola. Special thanks are also given to both Spanish Research Stations crews for their help in the summer cruises 2017–2019, as well as the ships'crews of Bio-Hespérides and Sarmiento de Gamboa for logistic support. We also acknowledge Quantarctica and the Norwegian Polar Institute for the use of the Quantarctica maps created for this paper. This study is part of the SCAR-Biology Programme–State of the Antarctic Ecosystem (AntEco: https://www.scar.org/science/anteco/home).

## Author Contributions

**Conceptualization:** Andrea M. Burfeid-Castellanos, Rafael P. Martín-Martín.

**Data curation:** Andrea M. Burfeid-Castellanos, Michael Kloster.

**Formal analysis:** Andrea M. Burfeid-Castellanos, Carlos Angulo-Preckler, Bánk Beszteri.

**Funding acquisition:** Andrea M. Burfeid-Castellanos, Conxita Avila.

**Investigation:** Rafael P. Martín-Martín, Conxita Avila, Bánk Beszteri.

**Methodology:** Andrea M. Burfeid-Castellanos, Michael Kloster.

**Project administration:** Andrea M. Burfeid-Castellanos, Conxita Avila.

**Resources:** Bánk Beszteri.

**Supervision:** Bánk Beszteri.

**Validation:** Andrea M. Burfeid-Castellanos, Carlos Angulo-Preckler.

**Writing – original draft:** Andrea M. Burfeid-Castellanos.

**Writing – review & editing:** Andrea M. Burfeid-Castellanos, Rafael P. Martín-Martín, Michael Kloster, Carlos Angulo-Preckler, Conxita Avila, Bánk Beszteri.

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
