## [Decision Letter · Decision Letter 0]

18 Feb 2021

PONE-D-20-37581

Epiphytic diatom community structure and richness is determined by macroalgal host and location in the South Shetland Islands (Antarctica)

PLOS ONE

Dear Dr. Burfeid-Castellanos,

Thank you for submitting your manuscript to PLOS ONE. After careful consideration, we feel that it has merit but does not fully meet PLOS ONE’s publication criteria as it currently stands. Therefore, we invite you to submit a revised version of the manuscript that addresses the points raised during the review process.

There are two parts to the revision.  Firstly, PlosOne only accepts methodological papers if there is a comparison between the old and new method- which you do not do.  Actually as you reference the method you use to Carr et al 1986 and Friedrichs 2013, I am not sure why you say this is a new method. Assuming that you can not in fact do this comparison I suggest all removal of the "new" method from the abstract.  This need not prevent you discussing the need for a consistent method in the discussion, and indeed to satisfy the first reviewer you do need to do this.  Here you should point out the possibility of you having missed some diatoms due to the bleaching process- although given that you appear to find higher diversity than expected this would make your results more marked.

Secondly, the reviewers both suggest places where clarification is necessary.  To that I add that you need to extend your statistical method section to discuss what you questions you actually used the packages (e.g., ANOSIM) to answer with what data.  It appears to me that you did a number of one way ANOSIMs which is fine- but actually stating this may solve reviewer 1's problem with the 26 samples and 20 states.

We look forward to receiving your revised manuscript.

Kind regards,

Judi Hewitt

Academic Editor

PLOS ONE

Journal Requirements:

2. We note that Figure 1 in your submission contains map images which may be copyrighted. All PLOS content is published under the Creative Commons Attribution License (CC BY 4.0), which means that the manuscript, images, and Supporting Information files will be freely available online, and any third party is permitted to access, download, copy, distribute, and use these materials in any way, even commercially, with proper attribution. For these reasons, we cannot publish previously copyrighted maps or satellite images created using proprietary data, such as Google software (Google Maps, Street View, and Earth). For more information, see our copyright guidelines: http://journals.plos.org/plosone/s/licenses-and-copyright.

(1) You may seek permission from the original copyright holder of Figure 1 to publish the content specifically under the CC BY 4.0 license. 

4. In your Methods section, please provide additional information regarding the permits you obtained for the work. Please ensure you have included the full name of the authority that approved the field site access and, if no permits were required, a brief statement explaining why.

Reviewers' comments:

Reviewer's Responses to Questions

**Comments to the Author**

1. Is the manuscript technically sound, and do the data support the conclusions?

Reviewer #1: No

Reviewer #2: Yes

2. Has the statistical analysis been performed appropriately and rigorously? 

Reviewer #1: No

Reviewer #2: Yes

3. Have the authors made all data underlying the findings in their manuscript fully available?

Reviewer #1: No

Reviewer #2: Yes

4. Is the manuscript presented in an intelligible fashion and written in standard English?

Reviewer #1: No

Reviewer #2: Yes

5. Review Comments to the Author

Reviewer #1: In this study, Burfeid-Castellanos et al. attempt to compare the diatom communities growing on various macroalgal hosts collected from the vicinity of two Antarctic islands located within the South Shetland Islands archipelago. The Authors then compare their results with those already published by other research groups. They also state that new, optimised methods have been designed and used to process the diatom samples and to analyse the obtained dataset and that these methods could improve the current practices and protocols for diatom analysis. Although I was thrilled to read about such findings and innovations, this submission is highly disappointing.

The manuscript is poorly written, and the language requires a thorough revision. Many of the sentences are confusing, and the various scientific terms are used incorrectly (check the more detailed comments listed below). A few examples of sentences that make it difficult to guess what the Authors were trying to say are listed here:

L111: "Ex situ, epiphytic diatoms were extracted using a small part of the macroalgae and centrifuging them in a known volume of water."

L112: "Diatoms were pre- and post-washed in distilled water centrifuging at 1000 rpm (Eppendorf centrifuge 5810 R, Eppendorf AG, Germany) for five minutes."

L205: "When considering that only one chlorophyte sample pertaining to a species are represented, most diatom taxa (those positioned intermediate between Phaeo- and Rhodophyte specificity) might be host generalists."

L315: "Observing only the macroalgal epiphytes, the numbers decreased to 120 species and 42 genera."

L369: In LI a higher variability in light impact could have been expected for the glacier inputs."

However, the main problem of this study is that the Authors seem to be unaware of several facts vital for the study design and interpretation of their results. Here I list a few of them:

1. Diatom frustules dissolve much faster in alcaic solutions than concentrated acids, and thus using bleach instead of acids is not a "more gentle" treatment. At least not for siliceous shells.

2. Centrifugation of a macroalgal piece is not sufficient to detach all diatoms from the macroalgal surface. If the Authors disagree, they should check the macroalgal pieces after centrifugation to ensure that all diatom cells were detached and that their obtained diatom samples were representative of the original community.

3. It is impossible to run a reliable statistical analysis to assess the influence of a variable with 20 states if the dataset contains 36 samples only. Similarly, the proposed dataset (that includes the previously published data from other Antarctic regions) does not allow to test for the "methodological bias" or "geographical effect." Those aspects could and should be mentioned and briefly discussed, but there are too few data and too many variables to run any meaningful statistical analysis. For example, the Authors do not consider factors such as seasonality (which is of paramount importance for the Antarctic organisms trying to make the most of the brief Antarctic summer), while the basic water parameters are not measured or assessed.

4. Macroalgal diversity is much lower around Continental Antarctica than the Maritime Antarctic Region, and thus it will be impossible to compare a larger number of macroalgal hosts from various Antarctic locations.

In summary, I am keen to see some new studies about diatoms in Antarctica as I am to read and learn about some more automated methods of diatom analysis. However, before this study can be published, the Authors need to rethink their approaches and interpretations. I hope that the detailed comments below can provide some constructive critique and ideas of how to deal with this subject.

Detailed comments:

Abstract: It is one of the most basic requirements for any kind of scientific report that the method used is both “repeatable” and “transparent” (.i.e. sufficiently well described). The fact that you have chosen a method that is different from those used in other studies is your choice, which is acceptable and may or may not be justified. But there is no reason to believe your method is more “repeatable” and more “transparent” than those used previously. Unless this is not what you meant. If so, rephrase.

Keywords: In nearly all of the currently known cases, diatom epiphytism on macroalgae is not a symbiosis understood as an interaction where the two organisms involved cannot live without each other (in general, epiphytic diatoms can thrive on other substrates while the macroalga will do just fine without any epiphytes). Although the definition of symbiosis has changed to some degree in the last years (many biologists will now accept the use of „symbiosis” for interactions where the two organisms do not require the presence of each other to thrive), „benthic symbiosis” may be a rather unfortunate and inaccurate term to describe the subject of your study. Consider replacing with „biofouling”, “benthic communities”, “benthic habitats” etc.

Keywords: to increase the searchability of this manuscript, you may want to replace “Deception and Livingstone island with “Deception Island” and “Livingstone Island”.

L66: A “shore” is dry land. A different word will be more appropriate.

L68: Why “on the other hand”?

L68: (“surface of […] in the Antarctic and Subantarctic regions”) This sentence suggests that in the Antarctic coastal habitats the diversity of substrates available for colonisation is poor, which is not exactly the case. Rephrase and be more specific.

L71: Rethink the meaning of “epiphyte”. “Epiphyte” can be a plant growing on another plant, but also any organisms growing on plants. The situation is even more complicated since algae are no longer members of the plants domain. Thus (this comment applies to other parts of this manuscript), macroalgae are not “macrophytes”. You should either not use a generic term “epiphytes” when the context can be confusing or explain what definition of “epiphytes” you are using. In the current sentence, “epiphytes” is not suitable as that includes both micro and macro-organisms and although they all “contribute to biofouling”, not all of them will “facilitate the adherence of other organisms”. Also, an epiphyte is a biofouler by definition, so it is not correct to say “epiphytes can contribute to biofouling”. Rephrase.

L72: “biofouling organisms” (not “agents”)

L73: (“those interactions”): Which “interactions”? “Those interactions” must refer to the previous sentence and you do not characterise interactions in the previous sentence.

L75: Check the meaning of “mutualism” and rephrase.

L77: I do not agree with this division. All of these studies provide both taxonomic and ecological information. It is also inappropriate to speak about “sets” of studies.

L81: One of the problems that affect the Authors reasoning throughout this manuscript is that they do not seem to be aware that macroalgal diversity is significantly lower in Continental Antarctica than the South Shetland Islands. Most of the species you have analysed from the latter location will not occur in the Ross Sea or the Davis Sea. Therefore, “sampling of broad host diversity with a biogeographic comparison” is impossible due to this basic fact. Keeping this in mind, the Authors should reformulate their various statements, suggestions, and conclusions.

L88: That both biotic and abiotic factors affect any living organism and communities is a well-known fact, not a “hypothesis”. Thus, you cannot “hypothesise”.

L94: What are those “macroscopically visible diatom community samples”? Explain.

L110: How were those diatoms “extracted” from the macroalgae? Centrifugation alone will not detach many of the firmly attached diatoms like Cocconeis or Planothidium.

L112: (“Diatoms were pre- and post-wasted…): ? This sentence is confusing. Rephrase.

L114: One of the highlights of this paper is the “new method”. Yet the method is described very poorly, and it is not clear either what the Authors actually did and what innovation they added. This whole section needs to be rewritten with some extra care.

L124: Why is it important to the reader that “for the resulting images 980 pixels equal 100 µm”. Do you refer to some images that will be published in the current paper? If so, cite those images here.

L128: (“2 GB file-size”): 2 GB

L130: Did you mean “for each slide”?

L133: This needs some more clarifications. How were these annotations made? Was the programme able to identify both valve and girdle views? What about broken valves? How were the teratologies distinguished? How many diatoms (%) were not identified? Surely, there must have been some yet undescribed species. Did anyone revise those automatic annotations? If so, how?

L141: What are those “textural malformations”? Can you actually observe something like that using LM?

L142: In epiphytic diatoms, a very large portion of malformations occurs simply due to the lack of space and crowding of quickly growing diatom assemblage.

L148 (here and elsewhere): revise how the reference is cited.

L152: I could not access these materials using the information cited. Revise so that any reader can easily find this data.

L155: How were the “branchedness” and “age” assessed? It is not explained.

L156: epiphytic diatom distribution

L164: “quite significant” suggests that something is less significant than just “significant”. This does not correspond to the numerical values with which you have linked those terms. Revise.

L165: Again, the link does not lead to any dataset.

L167: I have never heard of “macroalgal sociology”. I believe such a term does not exist. What did you mean?

L169: How did you assess the “degree of epiphytic diatom colonization”? To be able to compare this characteristic between various samples, you would need to know the surface from which all diatoms present in your sample were collected. Your methodology description does not indicate that such measurements were made.

L170: How can anything be “partially predominant”?

L171: “Navicula cf. perminuta” implies that you found a species that resembles N. perminuta. Since you do not know what species it actually is, you cannot say this species is a “generalist”.

L175: How can you recognize “textural teratologies” in digested material using LM?

L186: You cannot reliably assess the influence of the algal host species if you have a dataset of 36 samples collected from 20 different species (=one variable with 20 states). This means that most of the states of the variable tested (algal species) will occur only once. Your results can never be statistically important.

L188: What is the point of adding “macroscopically visible diatoms” to this dataset? According to Table 1, those diatoms were colonies of Berkeleya rutilans. Most likely they were attached to some benthic substrate and thus were part of the biofilm growing on that substrate. They should not be treated as substrate themselves as there may be diatoms growing on larger diatom taxa within your other samples as well. Thus, the treatment of your samples is not consistent.

L204: Having just one sample of chlorophytes, you are unable to say anything about diatom communities on chlorophytes.

205: ? I do not understand what you are trying to say. Is this a figure description? If so, cite the figure at the end of this sentence. “Might be host generalists” is already an interpretation and as such it is unsuitable for the “Results” section.

L216: How did you assess the “branching pattern” and “annuality”? Do these terms correspond to the previously used “branchedness” and “age”?

L250 How can anything “cluster together significantly”? Rephrase.

L254 (“This is further in line…): Revise this sentence.

L257: If there were sample with a low concentration of diatoms, why did you not use larger pieces of your macroalgae?

L265: How about “were found in samples collected from shallower locations”?

L278: How can a rarefaction curve be “oversaturated”? Your Fig. S2 does not show “oversaturated rarefaction curves”.

L283: You did not measure “average abundance per host leaf”.

L288: This study does not say anything about diatom “diversification”. Choose the correct term.

290: …was the most diversified?

L300: It is impossible to assess any “geographic effect” with so few samples and so many variables.

L318: It may be a revelation for you, but your method is by no mean “gentle”. Diatoms dissolve much faster in alkaline solution (like the bleach you have used) than in very strong acids. Thus, 10 min in boiling concentrated acids is more “gentle” to diatom frustules than 45 min in bleach. That is why ancient diatom frustules are very rarely preserved in alkaline sediments. There is a vast body of references explaining this phenomenon.

L318: High nutrient coastal habitats are not uncommon in continental Antarctica due to, among others, penguin rookeries that are often present near Antarctic stations and various sampling sites. It is not a characteristic that is typical of Antarctic Peninsula only.

L321: Did you mean “niche diversity”?

L324: You can only say that something is “reduced” if you know it used to be higher previously and then became reduced by some sort of a reducing factor. How about “remained” low?

L327: Which “numerous previous studies”? You stated that similar studies are very scarce elsewhere.

L338: Your study did not indicate any species-specificity. Your study design did not allow you to explore this issue.

L347: What are those “morphological and life history traits”? You did not explain how those traits were identified and classified.

L351: What is “host form”? Do not just make up terms.

L352: How about grazing, thallus shedding, abrasion, wave action etc.? There is virtually no habitat in which any microbial community would remain in “near stable […] proportions”. Thee is a lot of valuable literature that will help you understand the biofilm dynamics.

L354: Have you ever seen “2D formations” of diatoms? Rephrase.

L364: According to this, both Palmaria decipiens and Plocamium would be classified as “branched”. And yet the morphology of these two taxa is very different.

L370: In any shallow-water coastal habitat micronutrients are hardly ever limiting. Moreover, bird guano not uncommon in coastal Antarctic habitats is a much better source of both macro and micronutrients than “volcanic exudation”.

L373-375: I do not understand what you are trying to say here. Please revise.

Reviewer #2: The present manuscript is an important contribution to Antarctic science and phycology, mainly marine, related to polar ecosystems. The data are robust and have the potential to be published in this journal.

I just miss a better detail of the collection methods, because thus, in addition to ensuring reproducibility, it also makes it possible for other researchers to use the same collection method, since in Antarctica the conditions of experimentation are decisive for the success of the research. There are two questions about the method, but not limited to the need for further details.

1. How were the collected samples stored? In trays, in ZIP-type bags, or other? This is important even in order to guarantee the non-cross contamination of samples of epiphytic diatoms.

2. What would be "known volume of water" used for the centrifugation process and obtaining the diatoms samples?

I ask you to clarify these two points.

In the rest of the manuscript, I believe that the data are very well presented and discussed, I would only request that the errors in the links of some of the cited references be corrected (the term "Error! Reference source not found" often appears in the manuscript, which made it difficult understanding at first moment).

I also suggest publishing the methods on platforms like Protocols IO or dryad, so it is well known for other authors to be able to use the same methods and compare their data in all areas of Antarctica.

6. PLOS authors have the option to publish the peer review history of their article (what does this mean?). If published, this will include your full peer review and any attached files.

Reviewer #1: No

Reviewer #2: **Yes: **Filipe Victoria

---

## [Author Response · Author response to Decision Letter 0]

31 Mar 2021

Dear Dr. Hewitt,

we would hereby like to re-submit our manuscript titled “Epiphytic diatom community structure and richness is determined by macroalgal host and location in the South Shetland Islands (Antarctica)” to Plos ONE.

The updated links to data and material, as well as code and data have now been added to the manuscript.

We would like to thank both reviewers for their constructive comments. Especially those by reviewer 1 drew our attention to a number of points which were not clearly or carefully enough formulated; we tried to address all of these points. Below we reply to each of them and / or specify which changes were made to the manuscript in response.

Editor: There are two parts to the revision. Firstly, PlosOne only accepts methodological papers if there is a comparison between the old and new method- which you do not do. Actually as you reference the method you use to Carr et al 1986 and Friedrichs 2013, I am not sure why you say this is a new method. Assuming that you can not in fact do this comparison I suggest all removal of the "new" method from the abstract. This need not prevent you discussing the need for a consistent method in the discussion, and indeed to satisfy the first reviewer you do need to do this. Here you should point out the possibility of you having missed some diatoms due to the bleaching process- although given that you appear to find higher diversity than expected this would make your results more marked.

• Thank you for drawing our attention to this, since it seems that we were not clear in the description of what was the new about the methods used. The preparation method, as you rightly say, is not new, only in the context of Antarctic epiphytic diatom studies. The virtual microscopy method applied is somewhat new, although we’re citing previous studies having developed the slide scanning methods. Reviewer 1 also seems to have misunderstood our methods in a very substantial way since their comments imply they thought we used an algorithmic / automated method for taxonomic identification. This is absolutely not the case, taxonomic identifications were done manually, only not sitting directly at the microscope, but observing high-resolution slide scans over a browser-based image display and annotation platform. The according part of material and methods has been carefully reworded to make this point clear. About the hypothesized effect of the bleach treatment, we agree with the reviewer that it is somewhat counter-intuitive that an alkaline treatment should be milder than an acidic one; however, two detailed studies which have looked at this indicate this clearly (Carr 1986 and Friedrichs 2013, cited under the numbers 24 and 25). The reason is probably a combination of the low concentration used (10x diluted household bleach) and the short incubation time (30-45 minutes). Besides clarifying these points, we also reworded the discussion to make it clear that we cannot causally link the treatment method to the observed higher richness, since we did not do a systematic comparison of both treatment methods on the same set of samples. This will indeed be an interesting task for an upcoming study. Until then, we have to agree with the reviewer that broadly applicable methodological recommendations must wait.

Editor: Secondly, the reviewers both suggest places where clarification is necessary. To that I add that you need to extend your statistical method section to discuss what you questions you actually used the packages (e.g., ANOSIM) to answer with what data. It appears to me that you did a number of one way ANOSIMs which is fine- but actually stating this may solve reviewer 1's problem with the 26 samples and 20 states..

• We have just eliminated all mentions to such an imbalanced comparison, and kept those based on a more solid background (macroalgal host class, sampling location) …

We would like to thank both anonymous reviewers, as well as you as editor, for engaging themselves in reviewing and editing this work, especially in times as stressful as these, with a pandemic thwarting all plans. We addressed all the issues brought up by the reviewers and corrected the manuscript where necessary. Thank you all for improving this manuscript.

Kind regards,

Andrea Burfeid-Castellanos, Rafael P. Martín-Martín, Michael Kloster, Carlos Angulo-Preckler, Conxita Avila and Bánk Beszteri

Response to reviewer 1:

We want to thank reviewer 1 for taking so much time in correcting and constructively critiquing our manuscript. We will follow a point for point response to the issues raised by him/her/them.

Reviewer #1: In this study, Burfeid-Castellanos et al. attempt to compare the diatom communities growing on various macroalgal hosts collected from the vicinity of two Antarctic islands located within the South Shetland Islands archipelago. The Authors then compare their results with those already published by other research groups. They also state that new, optimised methods have been designed and used to process the diatom samples and to analyse the obtained dataset and that these methods could improve the current practices and protocols for diatom analysis. Although I was thrilled to read about such findings and innovations, this submission is highly disappointing.

The manuscript is poorly written, and the language requires a thorough revision. 

• English has been revised along the manuscript

Many of the sentences are confusing, and the various scientific terms are used incorrectly (check the more detailed comments listed below). A few examples of sentences that make it difficult to guess what the Authors were trying to say are listed here:

- L111: "Ex situ, epiphytic diatoms were extracted using a small part of the macroalgae and centrifuging them in a known volume of water."

• Changed to: “Epiphytic diatoms were extracted under laboratory conditions using a small part of the macroalgae, an overall appraisal of epiphyte incidence was made before scraping the surface into the receptacle with 80 - 100 ml of water depending on epiphyte density. The algae were also inmmersed and the samples were centrifuged [e.g. 23].”

- L112: "Diatoms were pre- and post-washed in distilled water centrifuging at 1000 rpm (Eppendorf centrifuge 5810 R, Eppendorf AG, Germany) for five minutes."

• Changed to: “Diatoms were washed in distilled water to reduce remaining salinity. Samples were homogenizedand centrifuged at 1000 rpm (Eppendorf centrifuge 5810 R, Eppendorf AG, Germany) for five minutes, and again filled up with deionized water. This procedure was repeated five times. Diatoms were prepared using the Friedrichs’ [24] variation of Carr et al.’s method [25], using ten times diluted bleach (Domol Hygiene Reiniger, AGB Rossmann GmbH), based on 5% sodium hypochlorite as the undiluted oxidizing agent, with a treatment period of 30-45 min depending on the amount of organic matter present in the sample. The thus cleaned diatom frustules were washed five times following the same procedure as before the bleach treatment.”

- L205: "When considering that only one chlorophyte sample pertaining to a species are represented, most diatom taxa (those positioned intermediate between Phaeo- and Rhodophyte specificity) might be host generalists."

• Changed to: “The ternary plot (Fig 3) further shows that most diatom taxa were shared amongst phaeo and rhodophytes. Since only one chlorophyte was sampled, some or all of these might represent host generalist taxa which would also be found on chlorophytes with more sampling effort.”

- L315: "Observing only the macroalgal epiphytes, the numbers decreased to 120 species and 42 genera."

• Changed to: “Even after eliminating the diatom samples from the dataset, a total of 120 species and 42 genera of epiphytic diatoms were identified on macroalgal samples, still surpassing the diversity found in previous studies”

- L369: In LI a higher variability in light impact could have been expected for the glacier inputs."

• Changed to: “In contrast, the light influx of LI does not depend on movement of lapilli, but varies due to the input from time constrained glacier melt [62], thus providing more stable irradiation for macroalgae and epiphytic diatoms during the sampled Antarctic summer.” 

However, the main problem of this study is that the Authors seem to be unaware of several facts vital for the study design and interpretation of their results. Here I list a few of them:

1. Diatom frustules dissolve much faster in alcaic solutions than concentrated acids, and thus using bleach instead of acids is not a "more gentle" treatment. At least not for siliceous shells. 

• Thank you for bringing this to our attention. Following to the study made by Lars Friedrichs (2012), the comparison showed that, counterintuitively, the bleaching method was gentler, i.e. preserved diatom silicate ultrastructures better, than the acid methods. Furthermore, the bleach method of diatom preparation was first described in 1986 and there the samples were subjected to 1h-4h bleach sessions on freshwater epiphytic diatoms by Carr and collaborators. Their research shows that diatoms can withstand the bleach preparation structurally sound, as long as neither high concentrations of bleach are used (5% NaClO in our study, 10x diluted, would be considered a low concentration) or the 4h of treatment are not surpassed (up to 45 minutes in our study should so. 

2. Centrifugation of a macroalgal piece is not sufficient to detach all diatoms from the macroalgal surface. If the Authors disagree, they should check the macroalgal pieces after centrifugation to ensure that all diatom cells were detached and that their obtained diatom samples were representative of the original community. 

• We have reworked this part to make it more understandable in the text (attached in the review points below). Macroalgal samples were initially scraped by hand and then shaken following Vettorato’s method (2010). Furthermore, we used a combination of methods (Chen et al 2012) in order to loosen up the adnate diatoms before centrifugation to assure the obtentionof natural ecological structure in diatoms, while preserving the macroalgal samples for further studies. Since most of these kinds of diatoms (species of Cocconeis and Achnanthes) were found in our samples, this seems to have sufficed. At this time, we are unable to further check the representability. 

3. It is impossible to run a reliable statistical analysis to assess the influence of a variable with 20 states if the dataset contains 36 samples only. Similarly, the proposed dataset (that includes the previously published data from other Antarctic regions) does not allow to test for the "methodological bias" or "geographical effect." Those aspects could and should be mentioned and briefly discussed, but there are too few data and too many variables to run any meaningful statistical analysis. For example, the Authors do not consider factors such as seasonality (which is of paramount importance for the Antarctic organisms trying to make the most of the brief Antarctic summer), while the basic water parameters are not measured or assessed.

• Thank you for this indication. Indeed, we had not taken seasonality into account and we have now added it into the discussion. Nonetheless, and with all of the faults of not having access to physicochemical traits of Antarctic waters at the time of sampling, as many of the other studies also did not record, we believe that the combined power of 192 samples taken in the seven studies, including ours, as described in supplement table 6 would be enough to gather a glimpse of the variation along the Antarctic continent.

4. Macroalgal diversity is much lower around Continental Antarctica than the Maritime Antarctic Region, and thus it will be impossible to compare a larger number of macroalgal hosts from various Antarctic locations.

• We have to respectfully disagree with this statement. The macroalgal richness displayed in Thomas and Jiang (1986) would contradict the statement. Also, studies on macroalgae like Gómez and Huovinen (2020) or Wiencke et al (2014) point out that the reduced biodiversity which is recorded in continental Antarctica could also be a bias due to predominant location of scientific stations in Maritime Antarctica. We believe that ice-free littorals of continental Antarctica have not been studied enough to claim that their diversity is significantly smaller than the Maritime coast, and genetical studies have shown genetic flux between some species of red algae are interlinked at least. Furthermore, probably due to climate change, some species of macroalgae are expanding and increasing the diversity on ice-free substrates (Guillemin et al. 2020, Dubrasquet et al. 2018, Wiencke et al 2014).

In summary, I am keen to see some new studies about diatoms in Antarctica as I am to read and learn about some more automated methods of diatom analysis. However, before this study can be published, the Authors need to rethink their approaches and interpretations. I hope that the detailed comments below can provide some constructive critique and ideas of how to deal with this subject.

Detailed comments:

Abstract: It is one of the most basic requirements for any kind of scientific report that the method used is both “repeatable” and “transparent” (.i.e. sufficiently well described). The fact that you have chosen a method that is different from those used in other studies is your choice, which is acceptable and may or may not be justified. But there is no reason to believe your method is more “repeatable” and more “transparent” than those used previously. Unless this is not what you meant. If so, rephrase.

• Thank you for mentioning this, of course we did not intend to doubt the transparency of the work of the cited literature. We have rewritten the phrase to: “We used a non-acidic method for diatom digestion, followed by slidescanning and diatom identification by manual annotation through a web-browser-based image annotation platform.”

Keywords: In nearly all of the currently known cases, diatom epiphytism on macroalgae is not a symbiosis understood as an interaction where the two organisms involved cannot live without each other (in general, epiphytic diatoms can thrive on other substrates while the macroalga will do just fine without any epiphytes). Although the definition of symbiosis has changed to some degree in the last years (many biologists will now accept the use of „symbiosis” for interactions where the two organisms do not require the presence of each other to thrive), „benthic symbiosis” may be a rather unfortunate and inaccurate term to describe the subject of your study. Consider replacing with „biofouling”, “benthic communities”, “benthic habitats” etc.

• We agree with the reviewer and we have changed the keyword to “biofouling”

Keywords: to increase the searchability of this manuscript, you may want to replace “Deception and Livingstone island with “Deception Island” and “Livingstone Island. 

• Changed as proposed by reviewer 1

L66: A “shore” is dry land. A different word will be more appropriate. ” 

• Changed to “coasts”

L68: Why “on the other hand”? 

• This particle was eliminated.

L68: (“surface of […] in the Antarctic and Subantarctic regions”) This sentence suggests that in the Antarctic coastal habitats the diversity of substrates available for colonisation is poor, which is not exactly the case. Rephrase and be more specific.

• Rephrased to say: The surface of macroalgal hosts also serves as habitat for benthic microalgae (mainly diatoms) in the Antarctic and Subantarctic regions.

L71: Rethink the meaning of “epiphyte”. “Epiphyte” can be a plant growing on another plant, but also any organisms growing on plants. The situation is even more complicated since algae are no longer members of the plants domain. Thus (this comment applies to other parts of this manuscript), macroalgae are not “macrophytes”. You should either not use a generic term “epiphytes” when the context can be confusing or explain what definition of “epiphytes” you are using. In the current sentence, “epiphytes” is not suitable as that includes both micro and macro-organisms and although they all “contribute to biofouling”, not all of them will “facilitate the adherence of other organisms”. Also, an epiphyte is a biofouler by definition, so it is not correct to say “epiphytes can contribute to biofouling”. Rephrase.

• Thanks for bringing this to our attention. Since previous publications have used the word “epiphytic” to also mean “habiting on macroalgae” (e. g. Majewska et al studies) we have attached the following sentence as explanation: “Although macroalgae cannot be interpreted either as synonymous with or part of plants in the systematic sense, these macroalgae-inhabiting diatom assemblages are usually referred to as “epiphytic” in the literature [2–4].”

L72: “biofouling organisms” (not “agents”) 

• Corrected and shortened sentence to “Interactions with surface-inhabiting diatoms can influence the performance of macro- and microalgae species ...”

L73: (“those interactions”): Which “interactions”? “Those interactions” must refer to the previous sentence and you do not characterise interactions in the previous sentence. 

• Corrected in (removed from) text: “Interactions with surface-inhabiting diatoms can influence the performance of macro- and microalgae species to acclimate or adapt to new ecosystem pressures like climate change [8,9] or invasive species from lower latitudes [10,11].”

L75: Check the meaning of “mutualism” and rephrase.

• Thank you for this comment. We have changed it to “macroalgae inhabiting diatoms”: “Previous studies on macroalgae- inhabiting diatoms have focussed on the taxonomic composition and ecology of the epiphytic diatom flora on macroalgae around Antarctica [2,3,5,12–15] or terrestrial habitats [16].”

L77: I do not agree with this division. All of these studies provide both taxonomic and ecological information. It is also inappropriate to speak about “sets” of studies.

• Although we agree there is no clear division between previous studies in this sense (they were all taxonomic and ecological), all of them differed in their breadth of taxonomic vs. geographic focus. We now rephrased these sentences to hopefully bring this idea across better. We have updated it to: “Some studies focussed on the floristics and ecology of taxonomically diverse hosts at a single location, constructing a flora of Vestfold Hills [12] and King George’s Island Potter Cove [2] respectively. Majewska et al. took a different approach by characterizing the epiphytic diatom ecology and flora of a small number of host taxa [3,13–15] across different locations [15].”

L81: One of the problems that affect the Authors reasoning throughout this manuscript is that they do not seem to be aware that macroalgal diversity is significantly lower in Continental Antarctica than the South Shetland Islands. Most of the species you have analysed from the latter location will not occur in the Ross Sea or the Davis Sea. Therefore, “sampling of broad host diversity with a biogeographic comparison” is impossible due to this basic fact. Keeping this in mind, the Authors should reformulate their various statements, suggestions, and conclusions.

• This sentence was now removed to avoid ambiguities; although we did not feel we were using these assumptions at this place.

L88: That both biotic and abiotic factors affect any living organism and communities is a well-known fact, not a “hypothesis”. Thus, you cannot “hypothesise”.

• Of course, we agree that this is a well-known fact, and think that much of macro-ecology is concerned with disentangling the concrete interactions of biotic and abiotic factors. We have rephrased it to: “Although limited sampling in this distant region affects our study and limits the causal interpretability of statistical comparisons, just as it does for similar investigations in general, we attempt to disentangle the correlative contribution of these factors to community differences, while also substantially extending our diatom floristic knowledge of the Antarctic region.”

L94: What are those “macroscopically visible diatom community samples”? Explain.

• Explained as: “macroscopically visible diatom community samples, i.e. diatom blooms visible to the naked eye, were taken”

L110: How were those diatoms “extracted” from the macroalgae? Centrifugation alone will not detach many of the firmly attached diatoms like Cocconeis or Planothidium.

• Macroalgal samples were previously scraped in 80-100 mL of water, inserted into it and shaken following to losen up the adnate diatoms before centrifugation, to assure the reflection of natural ecological structure in diatoms. Since most of these kinds of diatoms (species of Cocconeis and Achnanthes) were found in our samples, this might have sufficed. This was updated in the manuscript to: “Epiphytic diatoms were extracted under laboratory conditions using a small part of the macroalgae, an overall appraisal of epiphyte incidence was made before scraping the surface into the receptacle with 80 - 100 ml of water depending on epiphyte density. The algae were also inmmersed and the samples were centrifuged [e.g. 23]. After this, the macroalgal part was extracted again for further use.”

L112: (“Diatoms were pre- and post-wasted…): ? This sentence is confusing. Rephrase.

• Rephrased to: “Diatoms were washed in distilled water to reduce remaining salinity. Samples were homogenizedand centrifuged at 1000 rpm (Eppendorf centrifuge 5810 R, Eppendorf AG, Germany) for five minutes, and again filled up with deionized water. This procedure was repeated five times. Diatoms were prepared using the Friedrichs’ [24] variation of Carr et al.’s method [25], using ten times diluted bleach (Domol Hygiene Reiniger, AGB Rossmann GmbH), based on 5% sodium hypochlorite as the undiluted oxidizing agent, with a treatment period of 30-45 min depending on the amount of organic matter present in the sample. The thus cleaned diatom frustules were washed five times following the same procedure as before the bleach treatment.”

L114: One of the highlights of this paper is the “new method”. Yet the method is described very poorly, and it is not clear either what the Authors actually did and what innovation they added. This whole section needs to be rewritten with some extra care.

• The new methodology mostly refers to the digital microscopy version, that is available at all times and has not yet been used on “epiphytic” diatoms. This may make the availability of diatom species images, identification and re-identification easier. For the rewritten version see L112 correction.

L124: Why is it important to the reader that “for the resulting images 980 pixels equal 100 µm”. Do you refer to some images that will be published in the current paper? If so, cite those images here.

• This is a good point. Crossreference added: For the resulting images 980 pixels equal 100 µm, (e. g. see fig. 2 and images available in PANGAEA: https://doi.pangaea.de/10.1594/PANGAEA.925913).

L128: (“2 GB file-size”): 2 GB 

• Corrected in the text as proposed by reviewer 1.

L130: Did you mean “for each slide”?

• Corrected in the text as proposed by reviewer 1.

L133: This needs some more clarifications. How were these annotations made? Was the programme able to identify both valve and girdle views? What about broken valves? How were the teratologies distinguished? How many diatoms (%) were not identified? Surely, there must have been some yet undescribed species. Did anyone revise those automatic annotations? If so, how?

• The identification was not automatic, but made manually. The annotations (or identification labels) were added to the image and are readily available for correction and update in taxonomic names. For more information on BIIGLE see Langenkemper et al 2017. All diatoms were identified by a person to the lowest available taxonomic degree. We rephrased for clarity: “Diatoms contained within these sections were identified manually until ca. 500 identified specimens were reached for each sample, which mostly was the case during analysing the first of the three virtual slide image segments.”

L141: What are those “textural malformations”? Can you actually observe something like that using LM? 

• Rephrased to: “For each diatom specimen identified in this procedure, also their position (valvar vs. pleural view) and the presence of teratologic deformations was recorded. Teratologies refer to malformations, i.e. deviations from usual species-specific outline form or valve pattern, that can occur as a result of biotic and abiotic stresses [35].“

L142: In epiphytic diatoms, a very large portion of malformations occurs simply due to the lack of space and crowding of quickly growing diatom assemblage.

• This fact is new to us and we have not found any literature to support this claim. Furthermore, in an extensive review by Falasco et al (2009), it is reported that the density of the biofilm and its thickness can reduce the impact of heavy metals or extraneous substances. 

L148 (here and elsewhere): revise how the reference is cited.

• Thank you for bringing this to our attention. The crossreferences were still on and links to figures got disfigured there where they still existed.

L152: I could not access these materials using the information cited. Revise so that any reader can easily find this data.

• As written to the editor, the PANGAEA doi was in production because of a backlog in PANGAEA. It now exists and has been updated.

L155: How were the “branchedness” and “age” assessed? It is not explained.

• Explanation added to the second paragraph in the Materials and Methods section: “The macroalagal attributes of branching pattern, based on thallus morphology, and age, meaning the annuality of macroalgae (annual, biannual or pluriannual) of each species, were ascertained according to literature e.g. [19,21,22].”

L156: epiphytic diatom distribution

• This was changed to: “The differential ternary graph showing species distributions of the epiphytic diatoms between three host classes (Phaeophytes, rhodophytes and chlorophytes) was made using the ‘ggtern’ package [40].“

L164: “quite significant” suggests that something is less significant than just “significant”. This does not correspond to the numerical values with which you have linked those terms. Revise.

• Changed to very significant

L165: Again, the link does not lead to any dataset.

• Link to the embargoed data attached. These will be released upon publication.

L167: I have never heard of “macroalgal sociology”. I believe such a term does not exist. What did you mean?

• Eliminated the term sociology from the subtitle and changed to “Epiphytic diatom floristics and ecology”

L169: How did you assess the “degree of epiphytic diatom colonization”? To be able to compare this characteristic between various samples, you would need to know the surface from which all diatoms present in your sample were collected. Your methodology description does not indicate that such measurements were made.

• Addition to the methods: “Epiphytic diatoms were extracted under laboratory conditions using a small part of the macroalgae, an overall appraisal of epiphyte incidence was made before scraping the surface into the receptacle with 80 - 100 ml of water depending on epiphyte density. The algae were also inmmersed and the samples were centrifuged [e.g. 23]. After this, the macroalgal part was extracted again for further use.”

L170: How can anything be “partially predominant”?

• Thank you for bringing this to our attention. We eliminated “partially” because it was based on the predominance on specific host macroalgae.

L171: “Navicula cf. perminuta” implies that you found a species that resembles N. perminuta. Since you do not know what species it actually is, you cannot say this species is a “generalist”.

• Changed to: “The most frequent and predominant species of diatoms found in association with macroalgae (Fig 2) were generalist diatoms such as Pseudogomphonema kamtschaticum (Grunow) Medlin (up to 25% relative abundance in a sample, present in all but one samples) or as yet undescribed species as Navicula cf. perminuta Grunow (up to 64% relative abundance in a sample, present in all samples) and Pseudogomphonema sp. 1 (up to 59 % relative abundance in a sample, present in 29 samples).”

L175: How can you recognize “textural teratologies” in digested material using LM?

• Changed to: “Teratological frustules accounted for 0 to 2.3% of the counted cells. Diatoms had more teratologies on rhodophytes (with an incidence of up to 2.4% of the sample and for 57.89% of all samples) than on phaeophytes (with an incidence of under 1%, in 32.89% of the samples).”

L186: You cannot reliably assess the influence of the algal host species if you have a dataset of 36 samples collected from 20 different species (=one variable with 20 states). This means that most of the states of the variable tested (algal species) will occur only once. Your results can never be statistically important.

• We agree that the identification of species level macroalgae-epiphytic diatom comparison is not statistically important, and thus, we have eliminated those statistical enumerations which refer to it. However, there are enough datapoints to make comparative studies in the host class category, which we maintain. For further studies we intend to only use those samples that have multiple specimens, but in this study, we have used multiple specimens of Delisea pulchra, Iridaea cordata, Desmarestia anceps, Himantothallus grandifolius, Gigartina skottsbergii, Gymnogongrus turquettii, and Palmaria decipiens, since we consider this to be a large enough pool for statistic importance.

L188: What is the point of adding “macroscopically visible diatoms” to this dataset? According to Table 1, those diatoms were colonies of Berkeleya rutilans. Most likely they were attached to some benthic substrate and thus were part of the biofilm growing on that substrate. They should not be treated as substrate themselves as there may be diatoms growing on larger diatom taxa within your other samples as well. Thus, the treatment of your samples is not consistent.

• This is not the case. The macroscopically visible diatoms, or diatom blooms, were not attached to macroalgal substrate, but they were sampled by the snorkelers. As becomes apparent in the subsequent sections, we did not use the macroscopically visible diatoms for further study, but only for a descriptive aim. The treatment of the rest of the samples was made consistently.

L204: Having just one sample of chlorophytes, you are unable to say anything about diatom communities on chlorophytes.

• The same as in the point above, chlorophytes were used for floristics analysis, further ecological studies were only made with the statistically relevant phaeophytes and rhodophytes

205: ? I do not understand what you are trying to say. Is this a figure description? If so, cite the figure at the end of this sentence. “Might be host generalists” is already an interpretation and as such it is unsuitable for the “Results” section.

• Changed to: “The ternary plot (Fig 3) further shows that most diatom taxa were shared amongst phaeo and rhodophytes. Since only one chlorophyte was sampled, some or all of these might represent host generalist taxa which would also be found on chlorophytes with more sampling effort.”

L216: How did you assess the “branching pattern” and “annuality”? Do these terms correspond to the previously used “branchedness” and “age”?

• Thank you for bringing this to our attention. We have homogenized the terms of branchedness and age to branching pattern and annuality throughout the manuscript.

L250 How can anything “cluster together significantly”? Rephrase.

• Removed from the text

L254 (“This is further in line…): Revise this sentence.

• Sentence revised to: “This was corroborated by the results of the Mantel test (geographical distance matrix vs. diatom communities, r = 0.299***).”

L257: If there were sample with a low concentration of diatoms, why did you not use larger pieces of your macroalgae?

• We did not use larger pieces of macroalgae because we wanted to try to keep the epiphyte colonization comparable, and thus used equivalent surface areas of the macroalgal samples.

L265: How about “were found in samples collected from shallower locations”?

• Thank you. We have taken your formulation on this.

L278: How can a rarefaction curve be “oversaturated”? Your Fig. S2 does not show “oversaturated rarefaction curves”.

• Thank you, we have changed oversaturated to saturated.

L283: You did not measure “average abundance per host leaf”.

• Thank you for bringing this to our attention. Changed to: “Breakdown of average dissimilarity between epiphytic diatoms in Deception and Livingston Island locations (SIMPER).”

L288: This study does not say anything about diatom “diversification”. Choose the correct term.

• Changed to diversity.

290: …was the most diversified?

• Changed to “diversified from the rest”.

L300: It is impossible to assess any “geographic effect” with so few samples and so many variables.

• Thank you for the indication. Even though the final subdivision into studies was summarised for clarity, we consider the number of samples (192 in total) to be sufficient to assess the “geographic effect” (see Supplement table S6).

L318: It may be a revelation for you, but your method is by no mean “gentle”. Diatoms dissolve much faster in alkaline solution (like the bleach you have used) than in very strong acids. Thus, 10 min in boiling concentrated acids is more “gentle” to diatom frustules than 45 min in bleach. That is why ancient diatom frustules are very rarely preserved in alkaline sediments. There is a vast body of references explaining this phenomenon.

• We understand that the preparation with diluted bleach seems counterintuitive at best. We have used both acid and diluted bleach to prepare diatoms previously, but not, of course in this study. The method described by Carr et al. (1986) shows that the 45 minutes in low concentration bleach should not present the effects that you mention. As explained in point 3 of your main concerns, this method with diluted bleach has long been used on marine culture diatoms and provided better results for ornamentation and structure observation (Friedrichs, 2014).

L318: High nutrient coastal habitats are not uncommon in continental Antarctica due to, among others, penguin rookeries that are often present near Antarctic stations and various sampling sites. It is not a characteristic that is typical of Antarctic Peninsula only.

• We concede that other locations along continental Antarctica have increased nutrients due to animal depositions. Nonetheless, as Zacher et al (2010) stated, the nitrate levels are higher in the Antarctic peninsula, and although reduced not limiting factors in the Antarctic continent. However, iron is limiting in the continent, not in the Antarctic Peninsula. Thus, we changed it to: “A partial explanation of high diatom species richness in Antarctic-Subantarctic marine benthos might be the unusually high nutrient concentrations (especially of nitrate) surrounding the Antarctic peninsula [22] in combination with higher iron levels [47].”

L321: Did you mean “niche diversity”?

• Changed to “niche diversity”

L324: You can only say that something is “reduced” if you know it used to be higher previously and then became reduced by some sort of a reducing factor. How about “remained” low?

• Changed to “remained low”

L327: Which “numerous previous studies”? You stated that similar studies are very scarce elsewhere.

• “Numerous” removed.

L338: Your study did not indicate any species-specificity. Your study design did not allow you to explore this issue.

• Phrase removed. Subsequent phrase changed to: “In our study however, the difference between both islands could reflect different taxonomic coverages of sampling, rather than genuine biogeographic signal: whereas most of the samples from LI were rhodophytes (>82%), in DI the proportions were 40% rhodophytes and 46% phaeophytes.”

L347: What are those “morphological and life history traits”? You did not explain how those traits were identified and classified.

• Thank you for this comment. We have attached the previously used label of “branch pattern and annuality” to it.

L351: What is “host form”? Do not just make up terms.

• Homogenized to “branch pattern and annuality”

L352: How about grazing, thallus shedding, abrasion, wave action etc.? There is virtually no habitat in which any microbial community would remain in “near stable […] proportions”. Thee is a lot of valuable literature that will help you understand the biofilm dynamics.

• Particle eliminated. It now reads: “As the host age increases, so does the colonization by a mature biofilm community [58]. Maturity of the biofilm, and thus organism position inside a polysaccharide casing, could also protect the community living in it, as shown in several heavy metal studies [59,60].”

L354: Have you ever seen “2D formations” of diatoms? Rephrase.

• Changed to “organism position inside a polysaccharide casing”. (s. L 352 review for full sentence).

L364: According to this, both Palmaria decipiens and Plocamium would be classified as “branched”. And yet the morphology of these two taxa is very different.

• Thanks for bringing this to our attention. Reviewer 1 is right, the morphology of Palmaria and Plocamium are very different and, as shown in Table 1, Palmaria decipiens is categorized as laminar, not branched as Plocamium cartilagineum. The use of Palmaria in this sentence is an honest mistake of the authors that slipped through. Thus, changed Palmaria decipiens to Plocamium cartilagineum.

L370: In any shallow-water coastal habitat micronutrients are hardly ever limiting. Moreover, bird guano not uncommon in coastal Antarctic habitats is a much better source of both macro and micronutrients than “volcanic exudation”.

• The main penguin colonies at DI are situated towards the external part of the island, while our samples come from the inside. We believe that the guano – addition of penguin rookeries is too diluted by the time it reaches the volcano caldera, so that, as explained by Bendia et al 2018, the micronutrients providing from the “volcanic exudation” should be more available. Also trace metals, such as iron, seem to be more bioavailable, as described by Deheyn et al. 20005 and Bendia 2018.

L373-375: I do not understand what you are trying to say here. Please revise.

• Changed to: “In contrast, the light influx of LI does not depend on movement of lapilli, but varies due to the input from time constrained glacier melt [62], thus providing more stable irradiation for macroalgae and epiphytic diatoms during the sampled Antarctic summer.”

Response to Reviewer 2:

We want to thank reviewer 2 for the time in revision and the useful annotations to improve the manuscript.

Reviewer #2: The present manuscript is an important contribution to Antarctic science and phycology, mainly marine, related to polar ecosystems. The data are robust and have the potential to be published in this journal.

I just miss a better detail of the collection methods, because thus, in addition to ensuring reproducibility, it also makes it possible for other researchers to use the same collection method, since in Antarctica the conditions of experimentation are decisive for the success of the research. There are two questions about the method, but not limited to the need for further details.

1. How were the collected samples stored? In trays, in ZIP-type bags, or other? This is important even in order to guarantee the non-cross contamination of samples of epiphytic diatoms.

• Dear reviewer 2, thank you for bringing this to our attention. We have updated the manuscript methods section to read to read: “Macroalgae were obtained simultaneously with other benthic organisms at each sampling spot and pooled together in 1L receptacles. At the wet lab, the specimens were separated by phylum and identified to lowest taxon, usually species, following literature [19,20]. The species samples of one sampling site and day were kept in separate zip-style bags and frozen at -20ºC until further processing at the University of Barcelona. The macroalagal attributes of branching pattern, based on thallus morphology, and age, meaning the annuality of macroalgae (annual, biannual or pluriannual) of each species, were ascertained according to literature e.g. [19,21,22]. Epiphytic diatoms were extracted under laboratory conditions using a small part of the macroalgae, an overall appraisal of epiphyte incidence was made before scraping the surface into the receptacle with 80 - 100 ml of water depending on epiphyte density. The algae were also inmmersed and the samples were centrifuged [e.g. 23]. After this, the macroalgal part was extracted again for further use. Depending on epiphyte concentration, several aliquots were made and fixated using ethanol. 

Diatoms were washed in distilled water to reduce remaining salinity. Samples were homogenizedand centrifuged at 1000 rpm (Eppendorf centrifuge 5810 R, Eppendorf AG, Germany) for five minutes, and again filled up with deionized water. This procedure was repeated five times. Diatoms were prepared using the Friedrichs’ [24] variation of Carr et al.’s method [25], using ten times diluted bleach (Domol Hygiene Reiniger, AGB Rossmann GmbH), based on 5% sodium hypochlorite as the undiluted oxidizing agent, with a treatment period of 30-45 min depending on the amount of organic matter present in the sample. The thus cleaned diatom frustules were washed five times following the same procedure as before the bleach treatment. The frustule suspensions were then dripped onto coverslips, left to dry, quality checked and mounted using Norland Optical Adhesive 61 (refraction index = 1.56, Norland Products Inc., Cranbury, New Jersey, US).”

2. What would be "known volume of water" used for the centrifugation process and obtaining the diatoms samples?

• Thank you for bringing this to our attention. We have updated it to read: “Epiphytic diatoms were extracted under laboratory conditions using a small part of the macroalgae, an overall appraisal of epiphyte incidence was made before scraping the surface into the receptacle with 80 - 100 ml of water depending on epiphyte density. The algae were also inmmersed and the samples were centrifuged [e.g. 23]. After this, the macroalgal part was extracted again for further use.”

I ask you to clarify these two points.

In the rest of the manuscript, I believe that the data are very well presented and discussed, I would only request that the errors in the links of some of the cited references be corrected (the term "Error! Reference source not found" often appears in the manuscript, which made it difficult understanding at first moment).

• Sorry for this problem. We had not observed that the hyperlinks were still crossreferenced for the supplementary material of the manuscript. It has been corrected now.

I also suggest publishing the methods on platforms like Protocols IO or dryad, so it is well known for other authors to be able to use the same methods and compare their data in all areas of Antarctica.

• Thanks for the suggestion, we agree. We are currently preparing a methods paper in which this will happen.

---

## [Editor Report · Decision Letter 1]

12 Apr 2021

Epiphytic diatom community structure and richness is determined by macroalgal host and location in the South Shetland Islands (Antarctica)

PONE-D-20-37581R1

Dear Dr. Burfeid-Castellanos,

We’re pleased to inform you that your manuscript has been judged scientifically suitable for publication and will be formally accepted for publication once it meets all outstanding technical requirements.

Kind regards,

Judi Hewitt

Academic Editor

PLOS ONE
---

## [Editor Report · Acceptance letter]

22 Apr 2021

PONE-D-20-37581R1 

Epiphytic diatom community structure and richness is determined by macroalgal host and location in the South Shetland Islands (Antarctica) 

Dear Dr. Burfeid-Castellanos:

I'm pleased to inform you that your manuscript has been deemed suitable for publication in PLOS ONE. Congratulations! Your manuscript is now with our production department. 

Kind regards, 

on behalf of

Dr. Judi Hewitt 

Academic Editor

PLOS ONE